_Article_

# Melanin-like nanoparticles slow cyst growth in ADPKD by dual inhibition of oxidative stress and CREB

Yongzhan Sun [ID] [1,2,9], Quan Zou [3,9], Huizheng Yu [1], Xiaoping Yi [1], Xudan Dou [1], Yu Yang [ID] [4], Zhiheng Liu [1,5], Hong Yang [6], Junya Jia [7], Yupeng Chen [ID] [1,4 ✉], Shao-Kai Sun [ID] [8 ✉] & Lirong Zhang [ID] [1 ✉]

## Abstract

Melanin-like nanoparticles (MNPs) have recently emerged as valuable agents in antioxidant therapy due to their excellent biocompatibility and potent capacity to scavenge various reactive oxygen species (ROS). However, previous studies have mainly focused on acute ROS-related diseases, leaving a knowledge gap regarding their potential in chronic conditions. Furthermore, apart from their well-established antioxidant effects, it remains unclear whether MNPs target other intracellular molecular pathways. In this study, we synthesized ultra-small poly-ethylene glycol-incorporated $Mn^{2+}$-chelated MNP (MMPP). We found that MMPP traversed the glomerular filtration barrier and specifically accumulated in renal tubules. Autosomal dominant polycystic kidney disease (ADPKD) is a chronic genetic disorder closely associated with increased oxidative stress and featured by the progressive enlargement of cysts originating from various segments of the renal tubules. Treatment with MMPP markedly attenuated oxidative stress levels, inhibited cyst growth, thereby improving renal function. Interestingly, we found that MMPP effectively inhibits a cyst-promoting gene program downstream of the cAMP-CREB pathway, a crucial signaling pathway implicated in ADPKD progression. Mechanistically, we observed that MMPP directly binds to the bZIP DNA-binding domain of CREB, leading to competitive inhibition of CREB's DNA binding ability and subsequent reduction in CREB target gene expression. In summary, our findings identify an intracellular target of MMPP and demonstrate its potential for treating ADPKD by simultaneously targeting oxidative stress and CREB tran-scriptional activity.

**Keywords** Melanin-like Nanoparticle; Reactive Oxygen Species; cAMP-CREB Pathway; ADPKD Therapy
**Subject Categories** Genetics, Gene Therapy & Genetic Disease; Urogenital System

See also: **L Cassina & A Boletta**

## Introduction

Autosomal dominant polycystic kidney disease (ADPKD) is a life-threatening monogenic disease affecting more than 12 million individuals worldwide. ADPKD is primarily caused by mutations in the _PKD1_ or _PKD2_ genes, which code for polycystin 1 or polycystin 2. This disease is characterized by an increasing number of fluid-filled cysts in the kidneys that enlarge over time. This leads to a progressive decline in renal function, with over 50% of patients developing renal failure by the age of 60 (Bergmann et al, 2018; Cornec-Le Gall et al, 2019; Luo et al, 2023; Torres and Harris, 2019).

Recent studies have shown that abnormal signaling pathways are involved in the progression of ADPKD, including abnormal elevation of cyclic adenosine monophosphate (cAMP) levels (Hanaoka and Guggino, 2000; Ong and Harris, 2015; Wang et al, 2022; Zhou and Torres, 2022), reactive oxygen species (ROS) accumulation (Ishimoto et al, 2017; Kahveci et al, 2020; Maser et al, 2002; Pellegrini et al, 2023), intracellular calcium imbalance (Mangolini et al, 2016), cellular complexity (Ding et al, 2021; Muto et al, 2022) and RNA abnormality (Lakhia et al, 2022; Weisser et al, 2023; Zhang et al, 2020b). cAMP can bind and activate PKA kinase activity, which phosphorylates substrates such as CREB. Phosphorylated CREB binds to the CRE sequence on its target genes and activates their expression (Dinevska et al, 2023; Pizzoni et al, 2023; Zhang et al, 2020a). Kakade and colleagues found that suppression of CREB reduced cyst dilation in vitro (Kakade et al, 2016). In our previous study, we found that the expression of CREB target genes in cystic epithelial cells is associated with the progression of ADPKD. Moreover, inhibition of CREB transcription activity effectively suppressed cyst growth in vivo (Liu et al, 2021). These studies highlight the potential therapeutic strategy of targeted inhibition of the cAMP-CREB pathway in the treatment of ADPKD.

Another key feature of ADPKD is mitochondrial dysfunction and the accumulation of excessive ROS (Ishimoto et al, 2017; Lin et al, 2018; Padovano et al, 2018; Pellegrini et al, 2023; Podrini et al, 2020; Schreiber et al, 2019). ROS homeostasis is crucial for maintaining physiological cellular functions. However, excessive

[1]The Province and Ministry Co-sponsored Collaborative Innovation Center for Medical Epigenetics, Key Laboratory of Immune Microenvironment and Disease (Ministry of Education), State Key Laboratory of Experimental Hematology, Department of Biochemistry and Molecular Biology, School of Basic Medical Sciences, Tianjin Medical University, Tianjin, China. [2]School of Biomedical Engineering and Technology, Tianjin Medical University, Tianjin, China. [3]Department of Radiology, The Second Hospital of Tianjin Medical University, Tianjin, China. [4]Department of Urology, Tianjin Institute of Urology, The Second Hospital of Tianjin Medical University, Tianjin, China. [5]Department of Toxicology and Sanitary Chemistry, School of Public Health, Tianjin Medical University, Tianjin, China. [6]Department of Pharmacology, School of Basic Medical Sciences, Tianjin Medical University, Tianjin, China. [7]Department of Nephrology, Tianjin Medical University General Hospital, Tianjin, China. [8]School of Medical Imaging, Tianjin Medical University, Tianjin, China. [9]These authors contributed equally: Yongzhan Sun, Quan Zou. ✉E-mail: ychen@tmu.edu.cn; shaokaisun@tmu.edu.cn; lzhang@tmu.edu.cn

ROS levels can be detrimental to cells (Guo et al, 2013; Milkovic et al, 2019). Mitochondria serve as the primary sites for both ROS production and scavenging. Mitochondrial dysfunction can lead to excessive ROS accumulation, thereby inducing oxidative stress and damaging cells (Guo et al, 2013; Juan et al, 2021). Our previous research has established a positive correlation between the levels of ROS in kidneys from ADPKD patients and the severity of the disease. Moreover, we have found that activation of the antioxidant transcription factor Nrf2 can effectively decrease ROS levels, leading to a delay in ADPKD progression (Lu et al, 2020). These findings indicate that interventions aimed at enhancing ROS clearance hold promise for slowing down ADPKD progression.

Melanin, as an endogenous biopolymer, possesses remarkable antioxidant properties, biocompatibility, and biodegradability with minimal side effects (Jiang et al, 2020; Lin et al, 2016; Qi et al, 2019). To harness the antioxidant capacity of melanin, artificial melanin-like nanoparticles (MNPs) have been developed (Jiang et al, 2020; Liu et al, 2020). Previous studies have demonstrated the efficacy of MNPs in treating various diseases associated with elevated ROS, including ischemic stroke, periodontal disease, acute lung injury, acute kidney injury, sepsis-induced myocardial injury and wound healing (Bako et al, 2022; Bao et al, 2018; Cao et al, 2023; Liu et al, 2023; Liu et al, 2017; Mavridi-Printezi et al, 2023; Sun et al, 2019a; Zhao et al, 2018). Notably, MNPs have also exhibited excellent chelating capability towards a variety of metal ions due to the presence of phenolic hydroxyl and amine groups on their surface, thereby endowing MNPs with imaging capacity (Nedaei and Delavari, 2018; Zeng et al, 2020). Previous research by Sun et al, as well as our own group, has led to the successful synthesis of polyethylene glycol-bound $Mn^{2+}$-chelated melanin (MMPP) nanoparticles. These nanoparticles are noted for their ultrasmall hydrodynamic size, superior physiological stability, remarkable antioxidative properties against various harmful ROS, and proven effectiveness in treating both acute kidney injury and sepsis-induced myocardial injury (Jiang et al, 2020; Liu et al, 2023; Sun et al, 2019a). Though MNPs, including MMPP, have shown promise as effective ROS scavengers for treating a wide range of acute ROS-related diseases, their potential in treating chronic disorders remain largely unknown. Moreover, similar to small molecule drugs, it is crucial to comprehend the molecular targets of MNPs to maximize their therapeutic effectiveness and ensure their safety. While the antioxidant properties of MNPs have been extensively studied and well established, their ability to target other intracellular signaling pathways and molecules is unclear.

Our research aims to address these gaps by investigating the therapeutic potential of MMPP in ADPKD. We demonstrate the inhibitory effect of MMPP on cyst growth in vitro, as well as its efficacy in suppressing cystogenesis when administered intravenously in ADPKD mice. Mechanistically, our findings reveal that MMPP exerts its therapeutic effects by reducing oxidative stress and maintaining redox homeostasis. Importantly, we found that MMPP directly interacts with CREB through the bZIP DNA-binding domain, leading to the disruption of the CREB-DNA complex and subsequent suppression of CREB's transcriptional activity.

# Results

## Preparation and biodistribution of ultra-small MMPP

The glomerular filtration barrier (GFB) consists of three main components: the fenestrated endothelium of the glomerular capillaries, the basement membrane, and the podocytes with their foot processes. The GFB acts as a "size filter" allowing particles below 10 nanometers (nm) passing through and being absorbed by renal tubular cells (Du et al, 2017; Kamaly et al, 2016). MNPs used in previous studies were typically larger than 20 nm, exceeding the kidney's filtration threshold and thus preventing their accumulation in renal tissues (Chu et al, 2016; Deng et al, 2019). To address this, Sun and colleagues developed an innovative approach to synthesize ultra-small $Mn^{2+}$-chelated MNPs through an effective coordination and self-assembly method (Sun et al, 2019a). This approach was successfully applied in delivering MNP to renal tubular cells and treating acute kidney injury (AKI). We adopted the same synthesis strategy in this study. As illustrated in Fig. 1A, we first prepared MMP nanoparticles by mixing $Mn^{2+}$, melanin powder, and polyvinylpyrrolidone (PVP) in a dimethylsulfoxide (DMSO)/$H_2O$ solution. Thiol-terminated polyethylene glycol (HS-PEG) was then incorporated into the MMP surface to increase its physiological stability, resulting in the formation of MMPP nanoparticles. The characteristic peaks of PVP and PEG all appeared in the spectrum of MMPP nanoparticles, demonstrating the successful functionalization of melanin with PVP and HS-PEG (Fig. 1B). Additionally, the zeta potential of MMPP nanoparticles was measured to be −5.47 mV.

We next evaluated the stability and degradation patterns of MMPP in diverse media. Over a two-week period, MMPP nanoparticles demonstrated excellent dispersion in $H_2O$, phosphate-buffered saline (PBS, pH = 7.4), saline, and Dulbecco's Modified Eagle Medium (DMEM), without observable aggregation (Appendix Fig. S1A). The hydrodynamic size remained consistent during this interval (Appendix Fig. S1B), indicating robust colloidal stability.

The size and the hydrodynamic size of MMPP were about 2.3 nm and 6 nm, respectively (Fig. 1C,D), suggesting that MMPP can transverse the GFB and might be taken by renal tubular cells. As PEG can be detected by specific antibodies, we then performed immunohistochemistry (IHC) to localize the PEG component of the MMPP to determine the biodistribution of MMPP in the kidney. As shown in Fig. 1E, MMPP primarily accumulated in the cortex, with some MMPP nanoparticles accumulating in the medulla. Additionally, IHC staining of consecutive kidney sections showed that the distribution of PEG was comparable to that of LRP2, a marker indicating proximal tubules. Furthermore, a subset of PEGs exhibited co-localization with AQP2, a marker for collecting ducts, whereas no co-localization was observed between PEGs and SYNPO, a marker for glomerular podocytes (Fig. 1F). These results suggest that MMPP can cross the GFB and undergo reabsorption in the proximal tubules and collecting ducts.

## MMPP inhibits cyst growth in 3D-MDCK and embryonic kidney models

We and other researchers have reported that ROS play crucial roles in cyst growth. Furthermore, it has been observed that the inhibition of ROS-induced oxidative stress markedly delays ADPKD progression (Lu et al, 2020). Due to the effective clearance of ROS by MMPP and its ability to be absorbed by renal tubular cells, we aimed to investigate whether MMPP can inhibit the growth of renal cysts. We first employed a 3D Madin-Darby canine kidney (MDCK) in vitro cyst growth model. MDCK cells were first

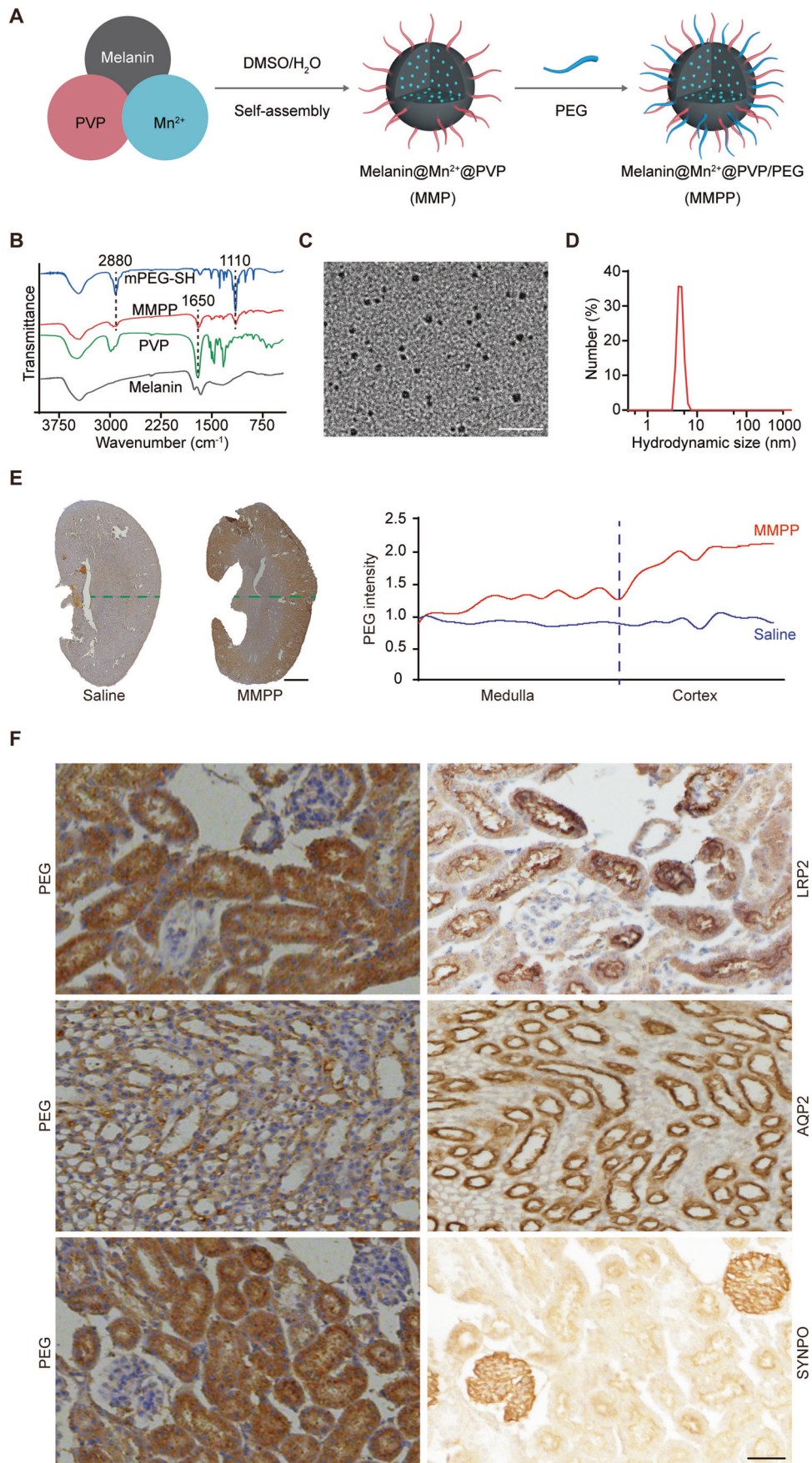

◄ **Figure 1.  The biodistribution of MMPP in mouse kidneys.**

(**A**) Schematic illustration of MMPP nanoparticles synthesis. (**B**) FTIR spectra of MMPP nanoparticles, HS-PEG molecule, PVP molecule and melanin granules. (**C**) TEM images of MMPP nanoparticles. (**D**) Hydrodynamic size of MMPP nanoparticles measured by DLS. (**E**) PEG staining (left) and signal intensity quantification (right) in whole kidneys with or without MMPP treatment. (**F**) PEG, LRP2, AQP2, and SYNPO staining in MMPP treated mouse kidney sections. Scale bars, 20 nm (**C**), 500 μm (**E**), and 100 μm (**F**). Source data are available online for this figure.

exposed to 10 μM forskolin (FSK), leading to the formation of cysts (Schreiber et al, 2019). Subsequently, MMPP was added in the culture medium from day 4 to day 12 following FSK treatment. As shown in Fig. EV1A, MMPP dose-dependently inhibited cyst growth. Furthermore, staining with Dihydroethidium (DHE) demonstrated that MMPP effectively reduced ROS levels in MDCK cysts in a dose-dependent manner (Fig. EV1B).

Next, we investigated the impact of MMPP on an ex vivo renal cyst model. Embryonic kidneys were harvested from CD1 mice at 13.5 embryonic days (E13.5) and cultured in transwells supplemented with 100 μM 8-Br-cAMP. MMPP was added to the transwell medium and incubated for a period of 6 days. Consistent with the MDCK model, MMPP exhibited a dose-dependent inhibition of embryonic renal cyst growth (Fig. EV1C). Remarkably, treatment with a high concentration of MMPP (400 μg/mL) resulted in an 80% reduction in cyst area (Fig. EV1C), while having minimal impact on the total renal volume (Appendix Fig. S2). Additionally, DHE staining also demonstrated a dose-dependent decrease in ROS levels in embryonic renal cysts following treatment with MMPP (Fig. EV1D). In summary, these findings indicate that MMPP effectively inhibits ROS production and hinders the growth of renal cysts both in vitro and ex vivo.

## MMPP delays cyst growth in ADPKD mouse model

It has been previously documented that intravenous (IV) injection of MMPP into mice leads to an accumulation of the nanoparticles in the kidneys, liver, and heart (Liu et al, 2023; Sun et al, 2019a). Sun et al demonstrated the use of $Mn^{2+}$ to enhance $T_1$-weighted magnetic resonance (MR) imaging, facilitating efficient visualization of in vivo circulation and organ distribution. Importantly, they also employed $Mn^{2+}$ tracing in an AKI mouse model to assess MMPP distribution under renal pathological conditions, providing a crucial reference for imaging-guided AKI therapy.

During the progression of ADPKD, a prominent feature is the disruption of normal kidney and liver structures by enlarging cysts (Torres et al, 2007). Therefore, ascertaining the in vivo distribution of MMPP in ADPKD, especially its efficacy in targeting the kidney and liver, was crucial. For detection the in vivo distribution of MMPP in wild type (WT) and ADPKD mice, we utilized MR imaging to monitor in vivo MMPP distribution using a 3T MR scanner over a 24-h period. As illustrated in Fig. EV2A, our findings aligned with those of Sun et al (Sun et al, 2019a), showing a preferential accumulation of MMPP in the kidneys and liver of normal mice. As shown in Fig. EV2B,C, we also observed a pronounced accumulation of MMPP in the liver and kidneys of ADPKD mice, with a slightly prolonged accumulation duration compared to normal mice.

We then investigated whether MMPP could inhibit cyst growth in ADPKD mice. The *Pkd1* knockout was induce by intraperitoneal (IP) injection of tamoxifen on postnatal days 25 (P25) and 28

(P28). The mice received IV treatment of MMPP (0.2 mg MMPP dissolved in 100 μL saline) once every 4 days from P55 for 2 months (Fig. 2A). As shown in Fig. 2B,C, MMPP treatment substantially reduced kidney size and kidney weight/body weight (KW/BW) ratio. Additionally, H&E staining revealed a marked decrease in cyst area and cystic index (Fig. 2D,E). Importantly, the renal function of ADPKD mice measured by blood urea nitrogen (BUN) (Fig. 2F) and creatinine (Appendix Fig. S3A) was substantially improved by MMPP treatment. Furthermore, we observed that MMPP treatment decreased kidney injury (Appendix Fig. S3B).

Liver cysts are the most common extrarenal manifestation of ADPKD (Chebib and Torres, 2016). Considering the accumulation of MMPP in the liver, we investigated the therapeutic potential of MMPP in treating liver cysts. As shown in Fig. 2G–I, MMPP treatment also reduced the number and volume of liver cysts in ADPKD mice. Taken together, these results suggest that MMPP effectively inhibits the growth of kidney and liver cysts in ADPKD mice.

We subsequently conducted a systematic safety analysis of IV injection of MMPP. No abnormal behavior and weight loss, were observed throughout the treatment. Interestingly, MMPP increased body weight in ADPKD mice (Fig. EV3A). Following a 2-month regimen of IV MMPP treatment in WT mice, primary organs including the kidney, liver, spleen, lung, and heart were collected for analysis. H&E staining of these organs showed no adverse effects in the MMPP group compared to the saline group (Fig. EV3B). Additionally, comprehensive hematologic analysis was conducted on blood samples from various administration groups. The results indicated that hematologic indices in MMPP-treated group were within normal ranges (Fig. EV3C). No significant differences in liver function measured by AST/ALT levels (Fig. EV3D,E). Taken together, these findings demonstrate that MMPP, when administered intravenously exhibit high safety profiles.

## MMPP inhibits oxidative stress by scavenging ROS in cystic kidney of ADPKD mice

To investigate the molecular mechanisms underlying the inhibitory effect of MMPP on cyst growth, we conducted transcriptomic analysis using RNA sequencing (RNA-seq) in collecting duct epithelial cells isolated from normal mice, ADPKD mice, and MMPP-treated ADPKD mice. We identified distinctive patterns of gene expression across different clusters in response to ADPKD onset and MMPP treatment. Specifically, genes in Clusters 1 and 5 did not show expression changes in ADPKD, but were either downregulated or upregulated following MMPP treatment. In contrast, genes in Clusters 3 and 6 experienced either upregulation or downregulation after ADPKD onset, yet remained unchanged following MMPP treatment. This behavior distinguished them from genes in Clusters 2 and 4, which undergo alterations

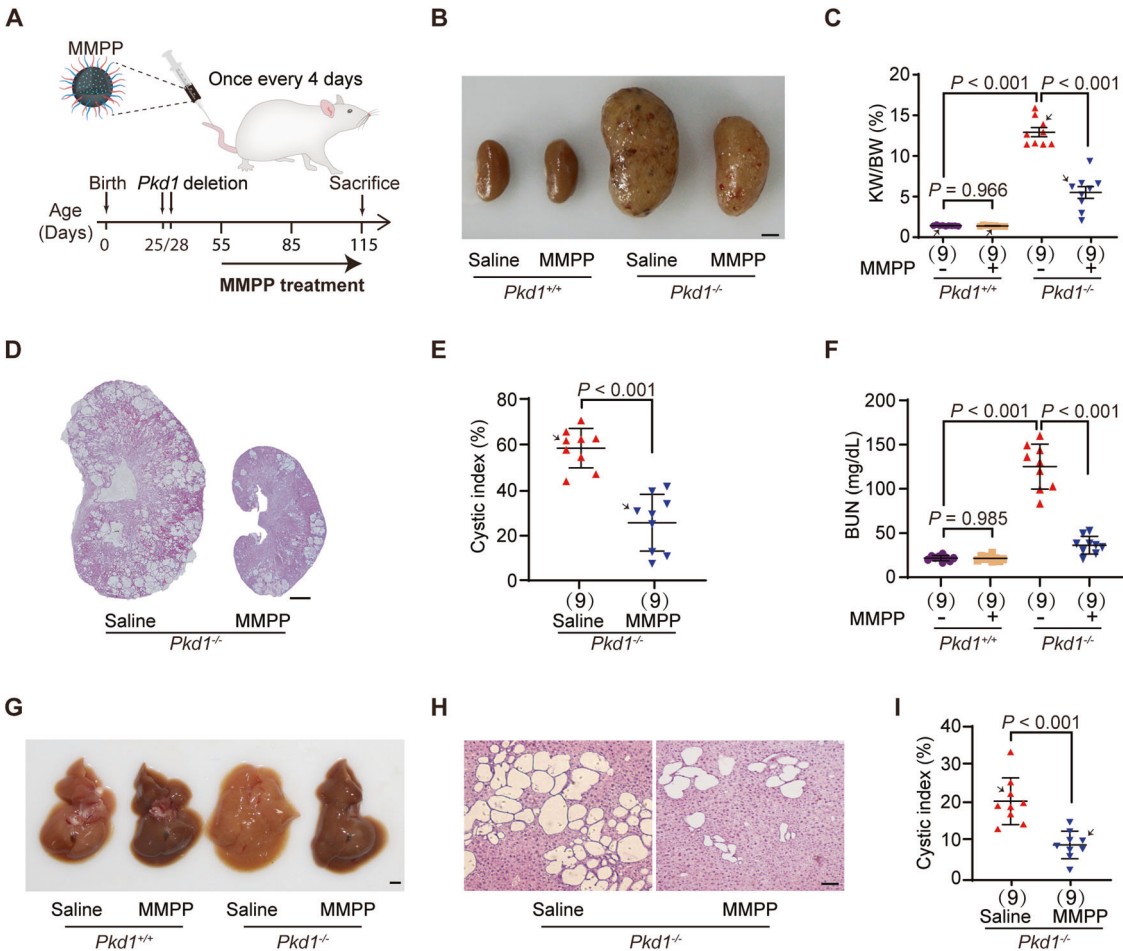

**Figure 2. MMPP attenuates cyst growth in ADPKD mice.**

(A) Schematic diagram of experimental design in ADPKD mouse model. (B) Representative images of kidneys from WT and $Pkd1^{-/-}$ mice treated with saline or MMPP. The representative images are from the data marked by black arrows in (C). (C) Ratios of kidney-weight to body-weight (KW/BW) in the indicated groups of mice ($n = 9$). (D) Hematoxylin and eosin (H&E) staining of kidney sections in saline- or MMPP-treated $Pkd1^{-/-}$ mice. The representative images are from the data marked by black arrows in (E). (E) Cystic index (cyst area to total area) calculated from H&E staining sections in (D) ($n = 9$). (F) Plasma BUN levels in mice from the indicated groups ($n = 9$). (G) Representative images of livers from WT and $Pkd1^{-/-}$ mice treated with saline or MMPP. (H) H&E staining of liver sections in saline- or MMPP-treated $Pkd1^{-/-}$ mice. The representative images are from the data marked by black arrows in (I). (I) Cystic index calculated from H&E staining sections in (H) ($n = 9$). Scale bars, 2 mm (B, D and G) and 100 µm (H). Data presented as means ± SD. Two-tailed unpaired Student's t test was used for statistical analysis in (E) and (I), two-way ANOVA with LSD test was used for statistical analysis in (C) and (F). Source data are available online for this figure.

post-ADPKD onset and were effectively restored by MMPP treatment, thus referred to as "MMPP-rescued genes" (Fig. 3A).

In ADPKD, the absence of PC1 led to structural and functional abnormalities in mitochondria (Ishimoto et al, 2017), disruptions in the tricarboxylic acid (TCA) cycle (Rowe et al, 2013) and a decrease in mitochondrial oxidative phosphorylation (OXPHOS) activity (Ishimoto et al, 2017). Dysregulation in amino acid metabolism (Soomro et al, 2018; Trott et al, 2018) and in lipid oxidation pathways (Lakhia et al, 2018) have also been observed. In the analysis of Cluster 2 genes, we performed GO enrichment analysis and observed a noteworthy enrichment in pathways linked to oxidative phosphorylation and metabolism, including oxidative stress, ATP production, and amino acid metabolism (Fig. 3B). These results indicate that MMPP plays pivotal roles in alleviating ADPKD progression. Consistently, gene localization analysis revealed that nearly one-quarter of the Cluster 2 genes were

located in mitochondria (Fig. 3C). We then validated the RNA-seq data by assessing the expression of several mitochondrial genes (*Co3*, *Cytb*, and *Atp6*) through quantitative real-time polymerase chain reaction (RT-PCR) analysis (Acin-Perez et al, 2004; Carr and Winge, 2003). As shown in Fig. 3D, their expression was effectively restored after MMPP treatment.

Since MMPP treatment effectively restores the expression of mitochondrial genes that are downregulated following ADPKD onset, we proceeded to examine the morphology and metabolic function of mitochondria in ADPKD kidneys post-MMPP treatment. We first utilized transmission electron microscopy (TEM) to investigate mitochondrial integrity. As shown in Fig. EV4A, mitochondria in ADPKD kidneys exhibited fragmented and abnormal shapes, with swelling and cristae damage—aligning with previous findings (Ishimoto et al, 2017; Lin et al, 2018). In contrast, in the MMPP-treated ADPKD kidneys, we noted that the mitochondria exhibited a morphology close

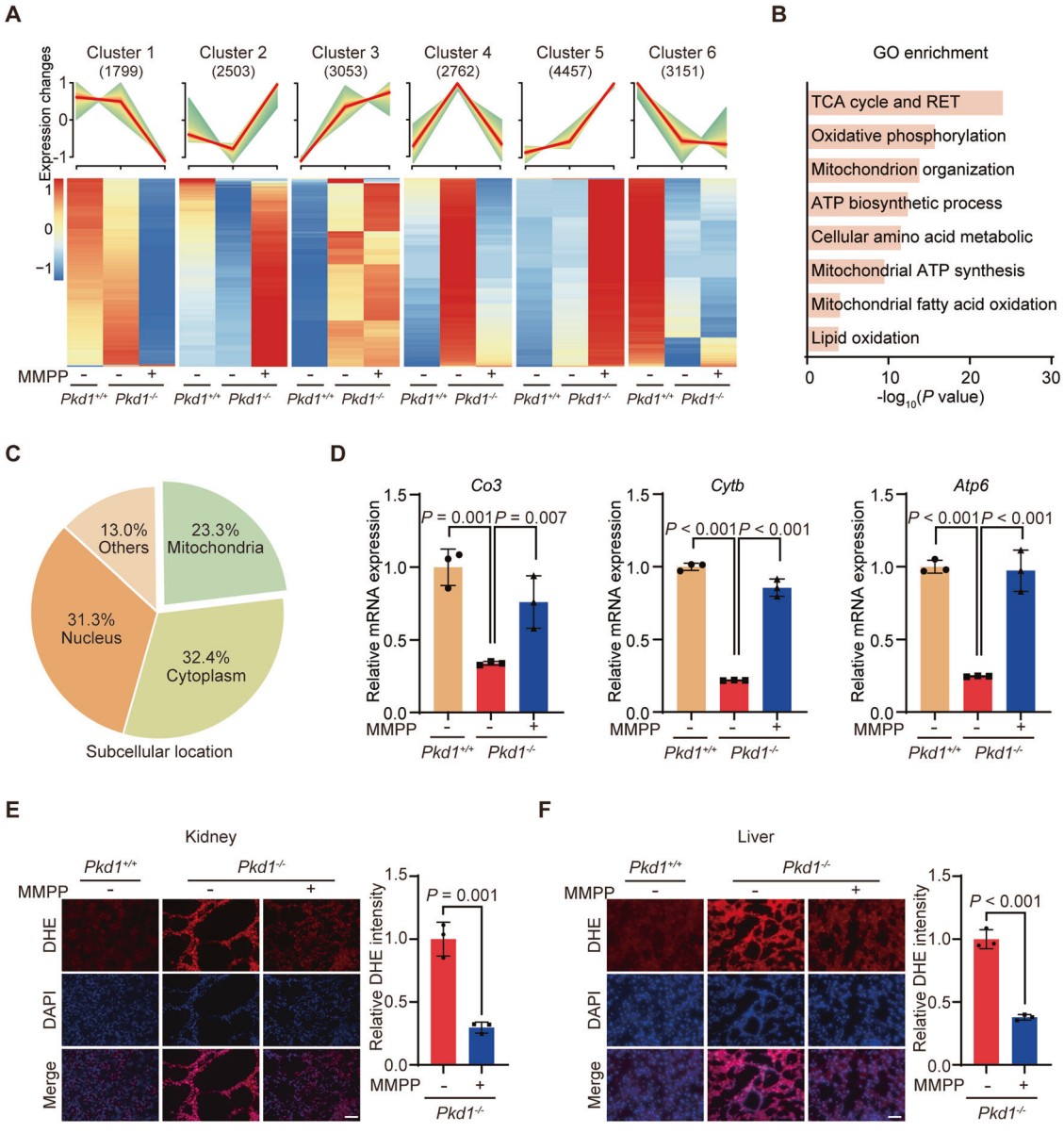

**Figure 3. MMPP inhibits oxidative stress by scavenging ROS in ADPKD mice.**

(A) Gene expression and clustering analysis of differentially expressed genes in cystic cells isolated from WT and $Pkd1^{-/-}$ mice treated with saline or MMPP. (B) Gene Ontology (GO) term enrichment analysis of Cluster 2 genes in (A). (C) Subcellular localization of the genes in Cluster 2. (D) RT-PCR analysis of the representative mitochondrial genes ($Co3$, $Cytb$ and $Atp6$) in primary renal epithelia cells isolated from the indicated groups of mice ($n = 3$). (E) DHE staining (left) and quantification (right) of frozen sections of mouse kidneys ($n = 3$). (F) DHE staining (left) and quantification (right) of frozen sections of mouse livers ($n = 3$). Scale bars, 50 μm (E, F). Data presented as means ± SD. Hypergeometric test was used for statistical analysis in (B), one-way ANOVA with LSD test was used for statistical analysis in (D), two-tailed unpaired Student's t test was used for statistical analysis in (E) and (F). Source data are available online for this figure.

to normal, characterized by reduced swelling and diminished injury to the cristae. Additionally, we isolated primary renal tubule cells from the respective mouse groups to measure their oxygen consumption rate (OCR). In line with existing literature (Ishimoto et al, 2017; Lin et al, 2018), primary cells from ADPKD kidneys showed a diminished OCR in comparison to those from WT kidneys, as depicted in Fig. EV4B. Notably, primary cells from MMPP-treated ADPKD kidneys exhibited an increase in both basal and maximal respiration rates, relative to primary cells from untreated ADPKD kidneys, as shown in Fig. EV4C,D.

To further investigate the in vivo ROS scavenging effect of MMPP, we collected kidney and liver samples and conducted ROS staining in ADPKD mice, both with and without MMPP treatment. As depicted in Fig. 3E,F, ROS levels were minimal in the kidneys and livers of normal mice, whereas they were prominently present in the tissues of ADPKD mice. Importantly, ROS staining in the kidneys and liver of ADPKD mice showed a marked reduction following MMPP treatment (Fig. 3E,F). Similarly, 8-hydroxy-2'-deoxyguanosine (8-OHdG) staining revealed that MMPP treatment decreased oxidative stress in ADPKD mice (Appendix Fig. S4).

Overall, these findings strongly indicate that MMPP treatment efficiently decreases the ROS level, restore mitochondrial morphology and metabolic functions in ADPKD mice through upregulation of crucial genes involved in maintaining oxidative balance and ROS homeostasis.

## MMPP decreases CREB target gene expression in cyst-lining epithelial cells

Transcription factors (TFs) play crucial roles in the regulation of gene expression. Our previous study demonstrated that the activation of CREB enhances cyst growth in ADPKD by upregulating the expression of genes associated with cystogenesis. Targeting CREB has shown promising therapeutic potential in animal models of ADPKD (Kakade et al, 2016; Liu et al, 2021). CREB regulates gene expression by binding to specific DNA sequences called cAMP response elements (CREs) (Conkright et al, 2003; Lonze and Ginty, 2002). Interestingly, transcriptomic analysis revealed that nearly one-third of Cluster 4 MMPP-rescued genes are CREB targets, which contain CRE sequences in their promoter and/or enhancer regions (Fig. 4A), suggesting MMPP might regulate CREB transcriptional activity. Interestingly, CREB target genes were more prominently enriched in a subset of MMPP-rescued Cluster 4 genes, which were highly expressed in ADPKD renal cells, suggesting their potential involvement in driving the progression of ADPKD (Fig. 4B).

Notably, the expression of these CREB targets was higher in ADPKD mice compared to normal mice, whereas their expression levels were effectively restored in ADPKD mice treated with MMPP (Fig. 4C). GO analysis revealed that MMPP-rescued CREB target genes in ADPKD were linked to key pathways such as RNA metabolism, ribosome biogenesis, amide metabolic processes, and immune system function (Fig. 4D). Furthermore, we examined the expression of several CREB target genes (*Myc*, *Sky*, *Abcg1*, and *Clcf1*), known for their involvement in ADPKD pathogenesis (Booij et al, 2017; Lee et al, 2020; Liu et al, 2019; Pandey et al, 2011). Consistent with the RNA-seq data, MMPP treatment effectively inhibited the upregulation of these CREB target genes in ADPKD (Fig. 4E).

In summary, these transcriptomic analyses suggest that MMPP inhibits ADPKD progression by exerting a dual effect. On one hand, it enhances the expression of mitochondrial functional genes, contributing to improved ROS clearing. On the other hand, it decreases the expression of CREB target genes, which are associated with ADPKD progression.

## MMPP inhibits CREB target genes expression through interacting with CREB

Next, we aimed to investigate the molecular mechanisms underlying the inhibitory effect of MMPP on CREB target genes expression. The inducible cAMP early repressor (*ICER*) is an endogenous repressor of cAMP-responsive element (CRE)-mediated gene transcription and belongs to CREB/CRE modulator (CREM)/ ATF-1 gene family (Borlikova and Endo, 2009; Lonze and Ginty, 2002), *NR4A2* belongs to the nuclear receptor 4 family of orphan nuclear receptors, CREB binds to its promoter with FSK treatment (Volakakis et al, 2010), both of which are classic CREB targets and the expression can be influenced by CREB

transcriptional activity. To address this, we first examined the effect of MMPP on the expression of CREB targets in vitro. As shown in Fig. 5A, FSK treatment markedly increased the expression of two CREB target genes, while pretreatment with MMPP dose-dependently suppressed their activation. We next tested the effect of MMPP in an immortalized *Pkd1*-null tubular epithelial cell line (PN) characterized by high endogenous cAMP levels (Sun et al, 2019b). Consistent with the results observed in 293T cells, the expression of *Icer* and *Nr4a2* was substantially reduced in MMPP-treated PN cells (Appendix Fig. S5). In addition, we observed a dose-dependent suppression of cAMP-induced CRE-luciferase reporter activity by MMPP treatment (Fig. 5B). These data suggest that MMPP can inhibit the expression of CREB target genes induced by high cAMP levels in cells cultured in vitro.

cAMP activates protein kinase A (PKA), which further phosphorylates CREB, promoting CREB's transcriptional activity (Screaton et al, 2004; Zhang et al, 2020a). We then tested whether MMPP affects the expression of CREB targets by modulating PKA activity. PKA activity can be evaluated using a phosphorylation antibody specific to PKA substrates, which recognizes the consensus sites of all PKA substrates (R(R/K)X(S*/T*)). As shown in Fig. 5C, FSK treatment resulted in a marked enhancement of PKA substrate phosphorylation. Interestingly, pre-treatment with MMPP did not impact this FSK-induced PKA substrate phosphorylation. The phosphorylation status of CREB is also crucial for the activation of its downstream targets. Notably, we observed a substantial increase in CREB phosphorylation following 1 h of FSK treatment, while MMPP pretreatment did not hinder this elevation of CREB phosphorylation (Fig. 5D). In summary, these findings indicate that MMPP does not inhibit the activation of CREB targets by modulating PKA activity or CREB phosphorylation.

The aforementioned results have prompted us to speculate on the possibility that MMPP suppresses CREB target genes by interacting with CREB. Previous studies have shown that MNPs with diameters larger than 80 nm predominantly localize in the cytoplasm (Kwon et al, 2022). Since CREB is a transcription factor that regulates gene transcription within the cell nucleus, we first assessed the intracellular distribution of our ultrasmall MMPP, particularly its ability to penetrate the cell nucleus. To test this, we modified MMPP nanoparticles by conjugating PEG with Biotin to create biotin-labeled MMPP nanoparticles (Bio-MMPP) (Fig. 5E). We then co-stained Bio-MMPP with various organelle markers—DAPI for nucleus, LAMP1 for lysosomes, TOM20 for mitochondria, and Calnexin for endoplasmic reticulum (ER). Our observations revealed that MMPP primarily localizes within the nucleus and lysosomes. Additionally, a fraction of MMPP was found to colocalize with mitochondria, whereas only minimal colocalization with ER was detected (Appendix Fig. S6).

To examine the interaction between Bio-MMPP and CREB, we overexpressed FLAG-CREB in 293T cells and treated them with 400 μg/mL of Bio-MMPP in the culture medium for 24 h. Next, the cells were harvested and cell lysate was pull-downed by streptavidin beads. As shown in Fig. 5F, a robust interaction between Bio-MMPP and FLAG-CREB was observed. We next examined the interaction of Bio-MMPP with endogenous CREB proteins by streptavidin beads pull-down, which revealed a robust interaction between Bio-MMPP and endogenous CREB proteins (Fig. 5G). We further conducted pull-down assays following cytoplasmic and nuclear fractionation in Bio-MMPP-treated 293T cells. We

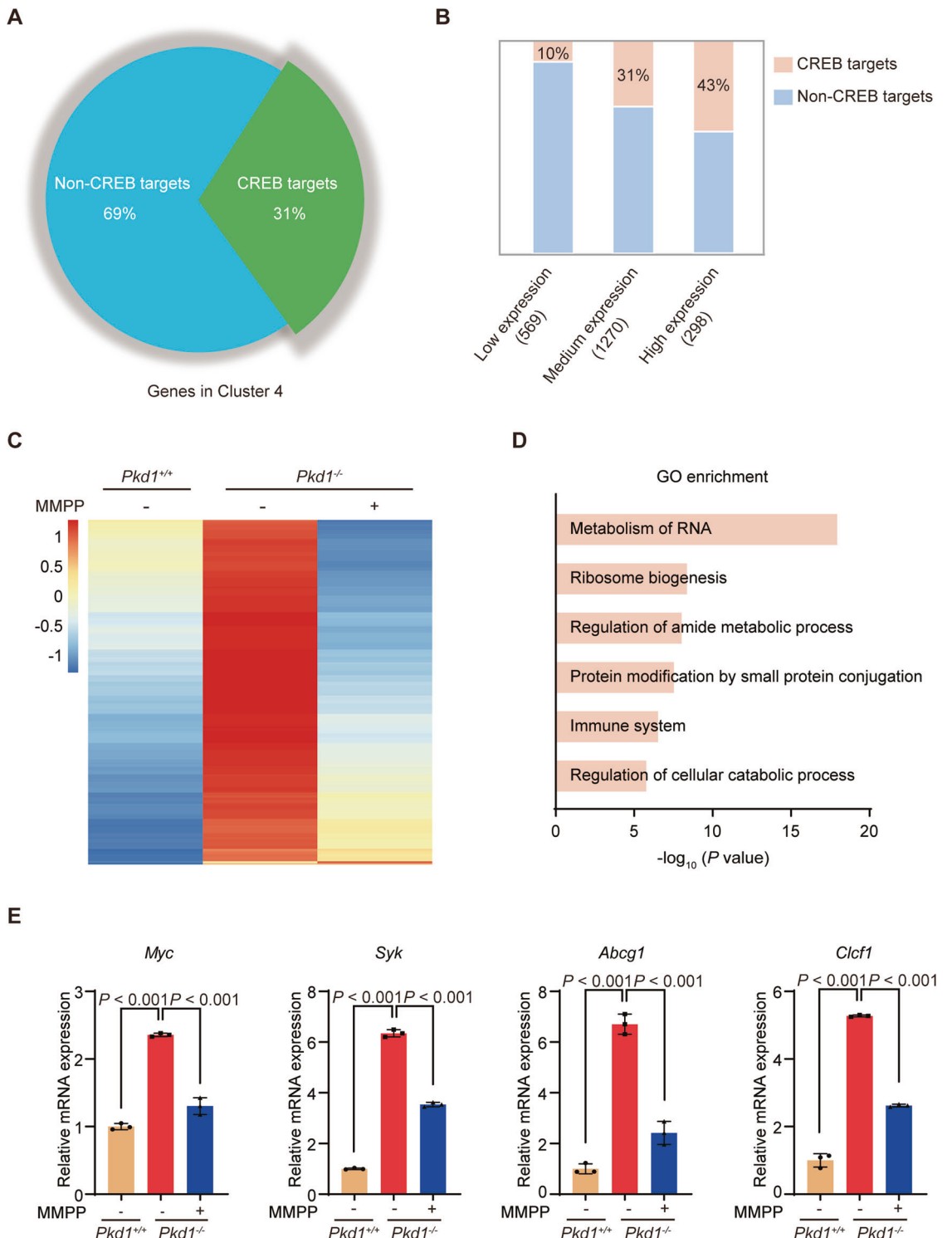

**Figure 4. MMPP inhibits the expression of CREB target genes in ADPKD mice.**

(A) Venn diagram illustrating the proportion of CREB target genes within Cluster 4 genes. (B) The proportions of CREB targets in genes with low, medium, or high expression level in ADPKD renal epithelial cells. (C) Heatmap displaying the expression levels of CREB targets within Cluster 4 genes. (D) GO term enrichment analysis of CREB target genes in (A). (E) RT-PCR analysis of representative CREB target genes (*Myc*, *Syk*, *Abcg1*, and *Clcf1*) in primary renal epithelia cells isolated from the indicated groups of mice (n = 3). Data presented as means ± SD. Hypergeometric test was used for statistical analysis in (D), one-way ANOVA with LSD test or Dunnett's T3 test was used for statistical analysis. Source data are available online for this figure.

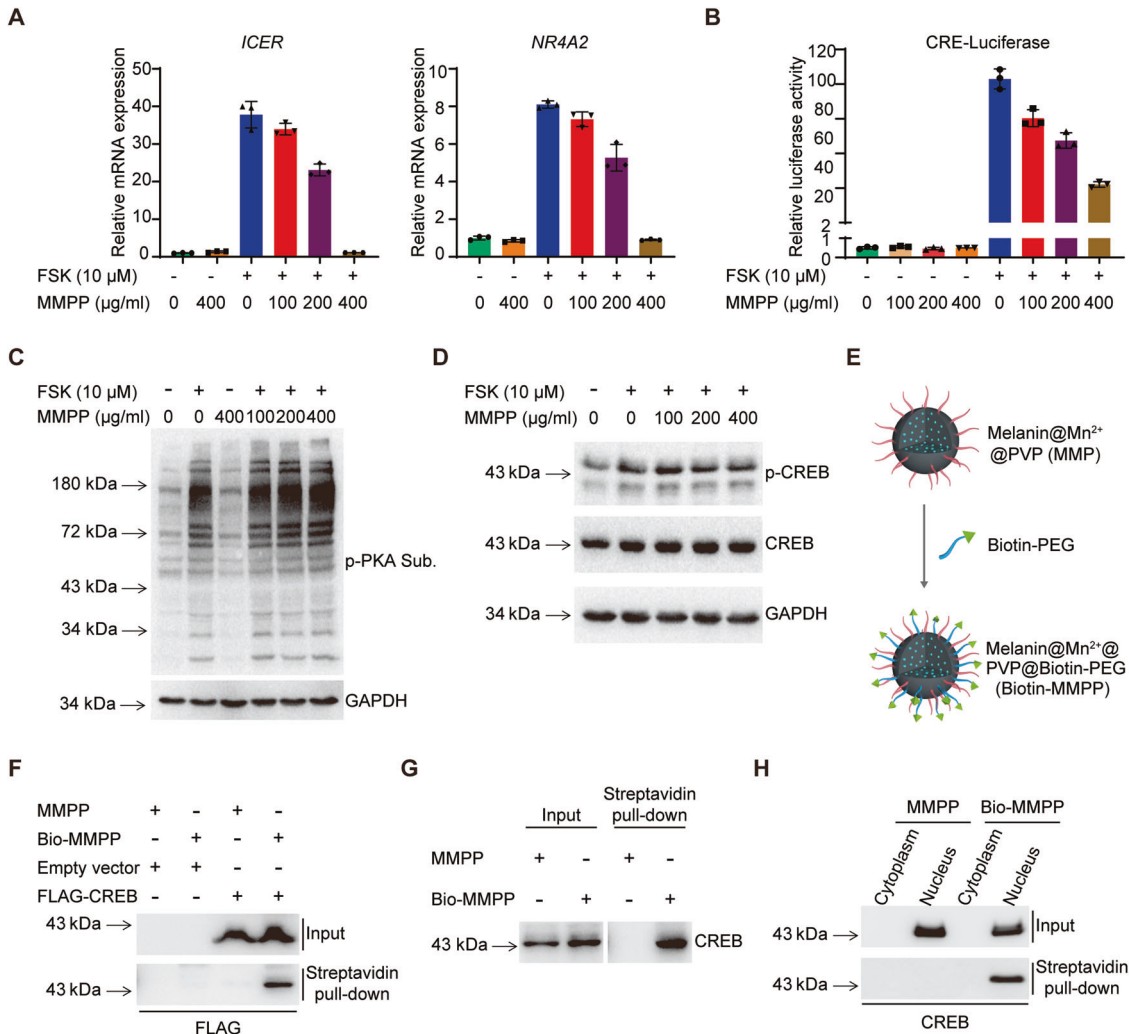

**Figure 5. MMPP inhibits CREB target gene expression through interacting with CREB.**

(A) RT-PCR analysis of representative CREB target genes (*ICER* and *NR4A2*) in 293T cells treated with the indicated doses of FSK and MMPP for 24 h (representative of three independent experiments; $n = 3$ biological replicates). (B) Quantification of CRE-luciferase activity in 293T cells treated with the indicated doses of FSK and MMPP for 24 h (representative of three independent experiments; $n = 3$ biological replicates). (C) Western blot analysis of PKA activity in 293T cells treated with the indicated doses of MMPP for 24 h, followed by 1-h FSK treatment. (D) Western blot analysis of p-CREB in 293 T cells treated as in (C). (E) Schematic illustration of Biotin-MMPP nanoparticle synthesis. (F) Streptavidin pull-down assays examining the interactions between Bio-MMPP (400 μg/mL) and FLAG-CREB in 293T cells. (G) Streptavidin pull-down assays examining the interactions between Bio-MMPP (400 μg/mL) and endogenous CREB in 293T cells. (H) Streptavidin pull-down assays examining the interactions between Bio-MMPP (400 μg/mL) and endogenous CREB in the cytosol and nuclear fractions of 293T cells. Data presented as means ± SD. Source data are available online for this figure.

observed a specific interaction between MMPP and endogenous CREB in the nuclear fraction (Fig. 5H), suggesting that MMPP may enter the nucleus and bind to CREB within the nucleus.

## MMPP directly interacts with CREB through the bZIP domain

To determine if there is a direct interaction between MMPP and CREB, we first purified the recombinant mCherry-CREB fusion protein from *E. coli* BL21 cells and generated Fluorescein isothiocyanate (FITC)-MMPP nanoparticles by conjugating FITC with PEG (Fig. 6A). Next, we added FITC-MMPP (100 μg/mL) to the solution containing 10% PEG 8000, which acts as a crowding

agent, with 40 μM purified mCherry-CREB or mCherry. As depicted in Fig. 6B, the mCherry-CREB protein, but not mCherry, exhibited droplet formation in the presence of crowding agents, and it colocalized with FITC-MMPP within these droplets. These observations indicate a direct interaction and colocalization between CREB and MMPP.

We then sought to identify the protein domain mediating the interaction between CREB and MMPP. The human CREB protein consists of four domains, including two glutamine-rich domains (Q1, amino acid 1–100 & Q2, amino acid 161–284), one kinase-inducible domain (KID, amino acid 101–160), and one leucine zipper domain (bZIP, amino acid 285-341) (Schumacher et al, 2000). We purified four truncated CREB proteins from *E. coli*

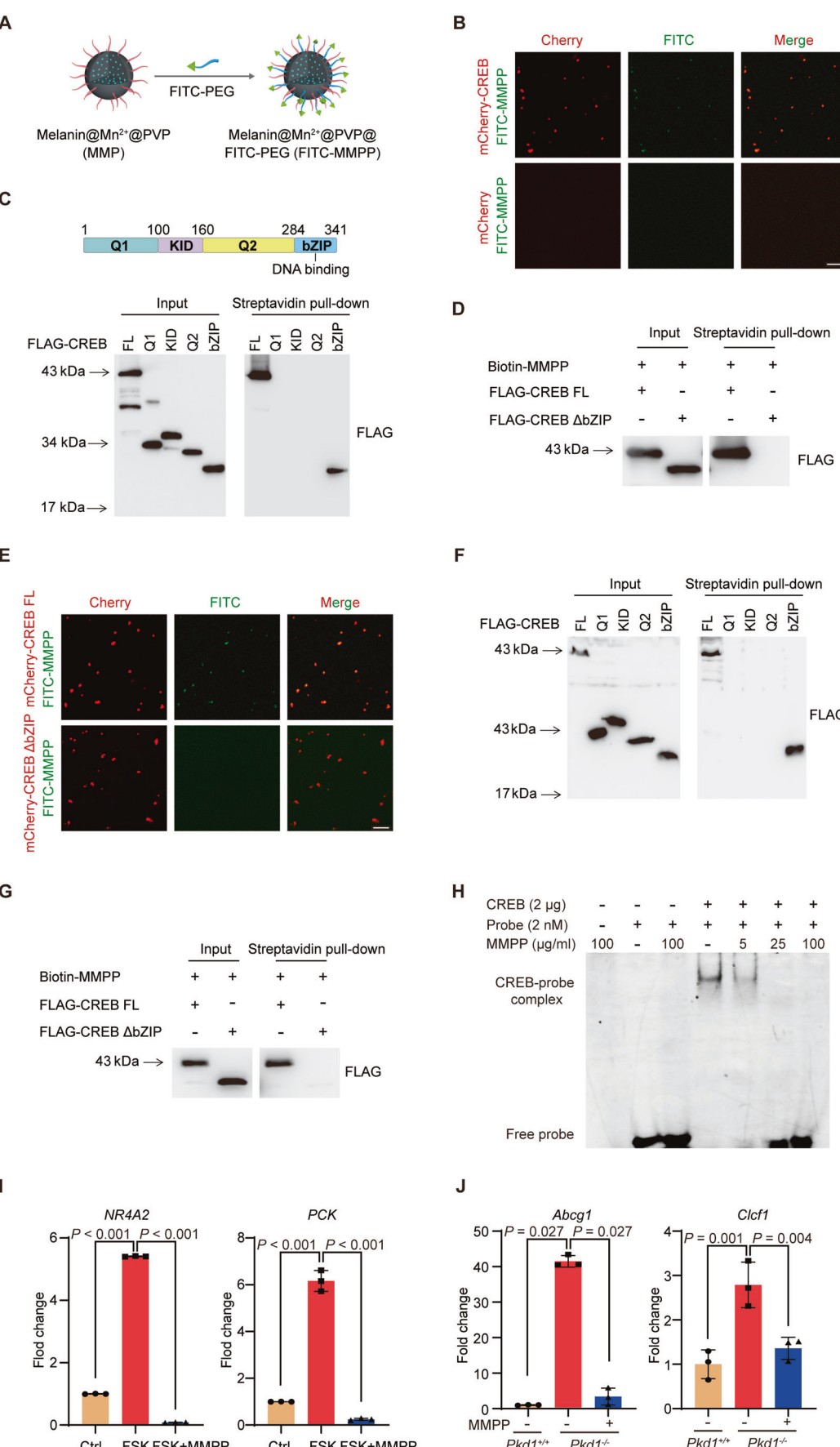

**Figure 6. MMPP directly interacts with CREB through the bZIP domain.**

(A) Schematic illustration of FITC-MMPP nanoparticle synthesis. (B) Representative images of in vitro colocalization analysis of purified Cherry/Cherry-CREB (40 µM) and FITC-MMPP nanoparticles (100 µg/mL). (C) Schematic diagram illustrating the functional domains of CREB (top) and Streptavidin pull-down assay (bottom) examining the interactions between Bio-MMPP (400 µg/mL) and purified CREB proteins containing full-length (FL) CREB, Q1 domain, KID domain, Q2 domain, or bZIP domain. (D) Streptavidin pull-down assay examining the interactions between Bio-MMPP (400 µg/mL) and purified full-length CREB or CREBΔbZIP truncated proteins. (E) Representative images of in vitro colocalization analysis of purified Cherry-CREB (40 µM)/Cherry-CREBΔbZIP (40 µM) and FITC-MMPP nanoparticles (100 µg/mL). (F) Streptavidin pull-down assays examining the interactions between Bio-MMPP (400 µg/mL) and full-length CREB, Q1 domain, KID domain, Q2 domain, or bZIP domain in 293T cells. (G) Streptavidin pull-down assays examining the interactions between Bio-MMPP (400 µg/mL) and full-length CREB or CREBΔbZIP overexpressed in 293T cells. (H) EMSA assays analyzing CREB binding to the CRE DNA probes with the indicated doses of MMPP treatment. (I) ChIP-qPCR analysis of p-CREB occupancy on *NR4A2* and *PCK* genes in 293T cells with indicated treatments (representative of three independent experiments; n = 3 biological replicates). (J) ChIP-qPCR analysis of p-CREB occupancy on *Abcg1* and *Clcf1* genes in primary renal epithelial cells isolated from WT and *Pkd1$^{-/-}$* mice treated with saline or MMPP (representative of three independent experiments; n = 3 biological replicates). Scale bars, 10 µm (B and E). Data presented as means ± SD. One-way ANOVA with LSD test was used for statistical analysis. Source data are available online for this figure.

BL21 cells, each containing a FLAG tag at the N-terminal. Bio-MMPP was then introduced into the solution containing purified full-length or truncated FLAG-CREB proteins, and MMPP pull-down assays were performed using Streptavidin beads. As shown in Fig. 6C, only the full-length CREB and bZIP domain exhibited an interaction with MMPP nanoparticles in vitro. Consistently, recombinant CREB proteins lacking the bZIP domain showed no interaction with MMPP (Fig. 6D). Similarly, the purified mCherry-CREB ΔbZIP proteins, while still capable of forming droplets, failed to recruit MMPP into these droplets (Fig. 6E).

To validate the interaction between MMPP and bZIP domain of CREB in mammalian cells, we overexpressed either full-length or truncated CREB in 293T cells and performed pull-down assays after Bio-MMPP treatment. As shown in Fig. 6F, only the bZIP domain of CREB interacted with MMPP in 293T cells, while other domains did not. Additionally, the CREB proteins lacking the bZIP domain (FLAG-CREB ΔbZIP) no longer interacted with MMPP (Fig. 6G). Taken together, these data suggest that the bZIP domain was critical for the direct interaction between CREB and MMPP both in vitro and in vivo.

## MMPP disrupts CREB DNA binding by competitively interacting with the bZIP domain

The bZIP domain of CREB binds CRE and is critical for the activation of CREB targets (Conkright et al, 2003; Schumacher et al, 2000). We thus hypothesized that MMPP may disrupt the CREB-CRE interaction by competing with CRE for bZIP interaction. To test this, we first synthesized a Cy3-labeled 30-nt double-stranded oligonucleotide probe (dsDNA) containing the CRE sequence. Next, we performed an electrophoretic mobility shift assay (EMSA) to determine the dsDNA binding affinity of recombinant CREB proteins in the present of MMPP. As shown in Fig. 6H, CREB exhibited robust binding to the CRE probes in the absence of MMPP. MMPP alone did not bind to the CRE probes; however, it dose-dependently reduced the binding affinity of CREB to the CRE probes. These findings indicate that MMPP disrupts the interaction between CREB and the CRE sequence in vitro.

To test whether MMPP blocks CREB-CRE binding in vivo, we examined the recruitment of p-CREB, the activated form of CREB, on CREB targets by performing a chromatin immunoprecipitation (ChIP) analysis of CREB in 293T cells with or without MMPP pre-treatment. As demonstrated in Fig. 6I, the recruitment of p-CREB on *NR4A2* and *PCK*, two classic CREB target genes, was significantly increased with FSK treatment but markedly decreased

in cells with MMPP pre-treatment. We then sought to investigate whether MMPP suppresses CREB transcription activity by disrupting CREB DNA binding in ADPKD mice. Consistent with the results obtained in 293T cells, the binding of p-CREB on the two selected CREB target genes, *Abcg1* and *Clcf1*, was markedly decreased in MMPP-treated ADPKD mice (Fig. 6J). Taken together, these data suggest that MMPP inhibits the expression of CREB targets by interfering the recruitment of activated CREB to CRE DNA element of CREB target genes.

## MMPP' inhibition on CREB transcriptional activity is independent of ROS scavenging

To determine if the inhibitory effect of MMPP on CREB is dependent on its general ROS scavenging activity, we conducted experiments using another two well-known antioxidants, sulforaphane (SFN) and N-acetylcysteine (NAC). SFN functions by activating the Nrf2 pathway, thus boosting the body's antioxidant defenses (Hong et al, 2010), while NAC acts to replenish glutathione levels, directly mitigating oxidative stress (Xie et al, 2018). The efficacy of these compounds in treating ADPKD has been previously established by our group and others (Lu et al, 2020; Moyses et al, 2016). WT 9-12 cells, which are derived from cysts of human PKD patients, exhibit elevated ROS levels compared to normal kidney collecting duct cells (Ishimoto et al, 2017). We conducted ChIP-qPCR assays on these cells treated with MMPP, SFN, or NAC. As shown in Appendix Fig. S7, there was a marked decrease in p-CREB recruitment on *NR4A2* and *PCK* genes in MMPP-treated cells. In contrast, such changes were not observed in cells treated with either SFN or NAC. These findings indicate that the inhibition of CREB binding to DNA by MMPP occurs independently of its ROS scavenging activity, highlighting the specificity of MMPP in targeting CREB function.

## Discussion

Owing to their excellent biocompatibility and potent capacity to scavenge a wide range of reactive oxygen and nitrogen species, MNPs have been extensively utilized in the antioxidant treatment of ROS-related diseases, including ischemic stroke, periodontal disease, and acute injuries affecting the myocardium, lungs, and kidneys (Bako et al, 2022; Bao et al, 2018; Cao et al, 2023; Kwon et al, 2022; Liu et al, 2023; Liu et al, 2017; Mavridi-Printezi et al, 2023; Sun et al, 2019a; Zhao et al, 2018). Nevertheless, the therapeutic potential of MNPs in

the context of chronic progressive diseases remains largely unexplored. One such chronic disorder is ADPKD, which is characterized by an extended disease course spanning several decades. In ADPKD, the principal clinical objectives are centered around enhancing renal function and impeding the progression of the disease (Borrego Utiel and Espinosa Hernandez, 2023; Zhou and Torres, 2023). Due to the loss of *PKD1/2* genes, cyst development can be initiated in the epithelial cells of different segments of the tubule (Bergmann et al, 2018; Calvet, 1998). To effectively traverse the kidney's filtration barrier and reach the renal tubular cells, we synthesized ultra-small MMPP with a diameter of 6 nm. Through in vivo biodistribution studies, we have observed their remarkable propensity to accumulate within renal tubular epithelial cells, particularly in proximal tubule cells. Importantly, administration of MMPP in ADPKD mouse models effectively suppresses oxidative stress, inhibits cystic epithelial cell growth, and leads to a notable improvement in renal function. Thus, this work broadens our understanding of MNPs in terms of their in vivo distribution patterns and therapeutic applications, thereby offering crucial insights for their prospective utilization in the treatment of diverse chronic ailments.

The identification and characterization of intracellular biomolecular targets of nanoparticles, similar to small molecular drugs, are crucial for understanding the biological mechanisms underlying their actions and developing safer and more effective nanoparticles, thereby facilitating future clinical translation (Augustine et al, 2020). However, whether MNPs can selectively target intracellular biomolecules and elicit specific biological responses, beyond their conventional antioxidant effects, remains elusive. Our previous research has demonstrated that inhibiting the transcriptional activity of CREB, a crucial downstream transcription factor of cAMP, effectively delays the progression of ADPKD by suppressing the expression of cyst-promoting genes (Liu et al, 2021). In the current study, we found that MMPP can directly interact with the bZIP domain of CREB, thereby competitively inhibiting the binding of CREB to target genes and suppressing CREB target gene expression. Therefore, MMPP exhibits promising therapeutic potential by simultaneously targeting two pivotal signaling pathways involved in ADPKD, namely oxidative stress and cAMP-CREB (Fig. 7). MMPP's unique mechanism of action, which not only scavenges ROS but also inhibits CREB transcriptional activity,

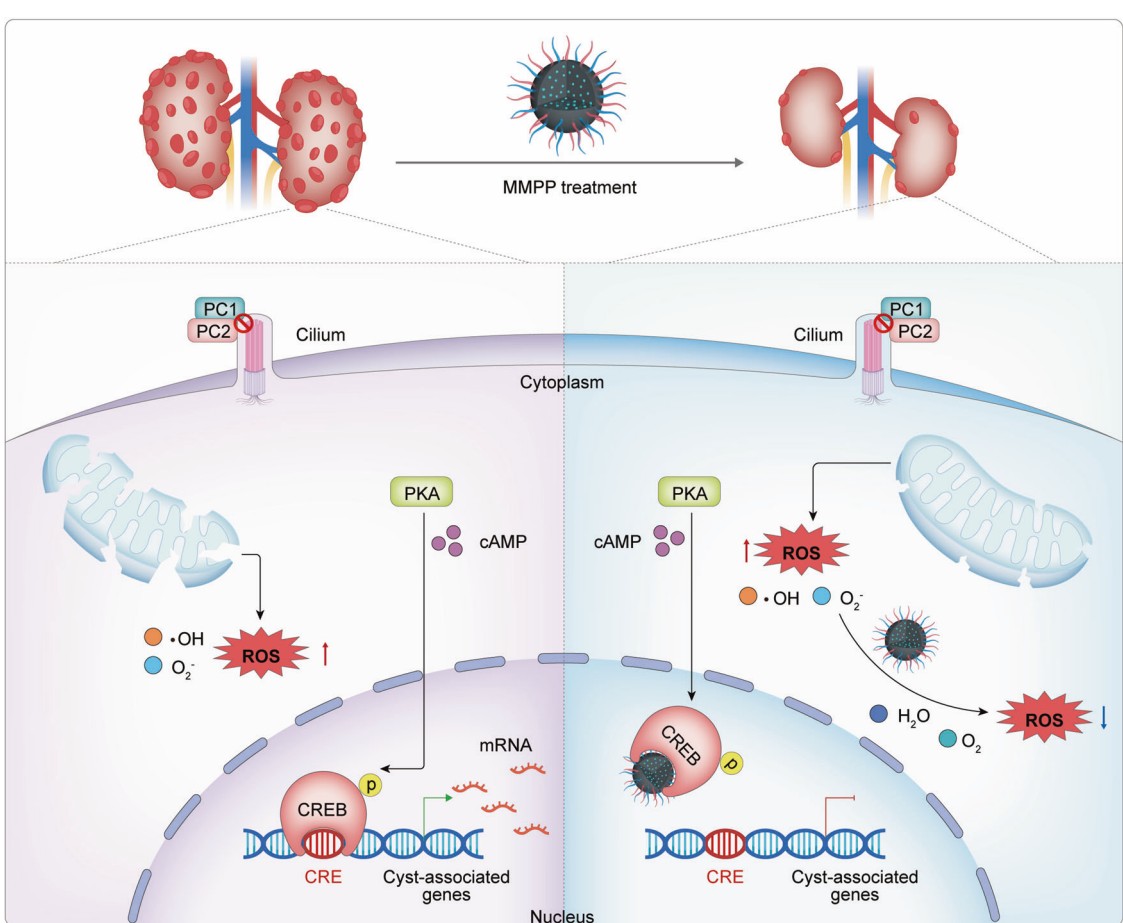

**Figure 7. Schematic illustration of the mechanism by which MMPP ameliorates ADPKD progression through dual inhibition.**

The left part shows that ROS accumulation and CREB hyperactivation accelerate ADPKD progression. The right part illustrates that MMPP treatment reduces ROS levels in the cytoplasm and inhibits CREB transactivation in the nucleus, thereby improving ADPKD progression. ROS reactive oxygen species, CRE cAMP response element, ·OH hydroxyl radical, $O_2^-$ superoxide anion radical.

particularly in the kidney and liver, suggests that MMPP could be a promising treatment for ADPKD.

The dysregulation of the cAMP-CREB signaling pathway is closely linked to the development of various diseases, such as cancer, neurological disorders, metabolic disorders and cardiovascular diseases (Bockus and Humphries, 2015; Dinevska et al, 2023; Ichiki, 2006; Wang et al, 2018; Zhang et al, 2020a), highlighting the potential of MNPs as a therapeutic approach for these conditions. However, it is important to recognize that the extensive regulatory influence of CREB across a wide array of genes and tissues raises the possibility of significant on-target adverse effects when utilizing drugs targeting CREB. While our thorough safety evaluations confirm a high safety profile, additional research using alternative ADPKD mouse models, notably those with slower progression such as the $Pkd1^{RC/RC}$ model, is essential to verify long-term effects and safety. Moreover, conducting studies in larger animal models, such as the porcine ADPKD model that more closely mirrors human physiology, will be instrumental in supporting its clinical application. Collectively, our study has expanded the understanding of the mechanism of action of MNPs and offered new prospects for developing safer and more effective MNPs with broader applications.

# Methods

**Reagents and tools table**

| Reagent/Resource | Reference or Source | Identifier or Catalog Number |
|---|---|---|
| **Experimental Models** | | |
| $Pkd1^{fl/fl};Cre/Esr1^+$ (M. musculus) | (Yang et al, 2018) | |
| Embryonic kidney (M. musculus) | (Sun et al, 2019b) | |
| 3D MDCK (Canis familiaris) | (Magenheimer et al, 2006) | |
| **Recombinant DNA** | | |
| pCREB | (Schumacher et al, 2000) | |
| pCREB-Q1 | (Schumacher et al, 2000) | |
| pCREB-KID | (Schumacher et al, 2000) | |
| pCREB-Q2 | (Schumacher et al, 2000) | |
| pCREB-bZIP | (Schumacher et al, 2000) | |
| **Antibodies (dilution)** | | |
| p-PKA substrate 1:1000 | Cell Signaling Technology | #9624 |
| CREB 1:1000 | Cell Signaling Technology | #9192 |
| p-CREB 1:1000 | Cell Signaling Technology | #9198 |
| FLAG-HRP 1:5000 | Sigma-Aldrich | #A8592 |
| 8-OHdG 1:200 | Bioss | #bs-1278R |

| Reagent/Resource | Reference or Source | Identifier or Catalog Number |
|---|---|---|
| PEG 1:50 | GenScript | #A01795 |
| AQP2 1:1000 | Proteintech | #29386-1-AP |
| SYNPO 1:200 | Proteintech | #21064-1-AP |
| LRP2 1:100 | ABclonal | #A3612 |
| **Oligonucleotides and other sequence-based reagents** | | |
| RT-PCR primers | This study | Appendix Table S1 |
| ChIP-qPCR primers | This study | Appendix Table S2 |
| **Software** | | |
| SPSS | www.ibm.com/spss | |

## Preparation of MMPP/Bio-MMPP/FITC-MMPP

The MMPP nanoparticles were synthesized following established procedures in (Sun et al, 2019a). Initially, 5 mg of melanin (Sigma-Aldrich, M8631) was dissolved in 5 mL of DMSO and stirred. Subsequently, 18 mg of $MnCl_2 \cdot 4H_2O$ and 50 mg of PVP in 5 mL of DMSO were added to the solution. After 12 h of stirring, the mixture was introduced to 100 mL of $H_2O$. The Melanin-Mn-PVP nanoparticles were then purified through centrifugal ultrafiltration. Following this, the pH of the Melanin-Mn-PVP solution was adjusted to 9.5 using an ammonium hydroxide solution. It was then combined with HS-PEG (5 mL, 10 mg/mL) or HS-PEG-Biotin (5 mL, 10 mg/mL) or HS-PEG-FITC (0.5 mL HS-PEG-FITC, 10 mg/mL + 4.5 mL HS-PEG, 10 mg/mL) under argon gas protection and stirred continuously for 12 h. Finally, the nanoparticles were purified through multiple rounds of centrifugal ultrafiltration (molecular weight cutoff 30 kDa) using deionized water.

## Cell culture

HEK293T cells (ATCC, CRL-3216), HK-2 cells (ATCC, CRL-2190), and WT 9-12 cells (Lu et al, 2020) were cultured in Dulbecco's modified Eagle's medium (DMEM) supplemented with 10% fetal bovine serum (FBS). Primary cells and *PKD1* null (PN) cells (Mi et al, 2020) were cultured in DMEM/F12 supplemented with 3% FBS and 0.75 μg/L γ-interferon (Sigma-Aldrich, I4777), 1 g/L insulin, 0.67 mg/L sodium selenite, 0.55 g/L transferrin (Thermo Fisher, 41400045), 0.1 μM 3,3,5-triido-l-thyronine (Sigma-Aldrich, T6397) and 1% penicillin–streptomycin. Cells obtained from ATCC were authenticated by SRT profile, all of the cells were tested for the presence of mycoplasma contamination and were maintained at 37 °C incubator containing 5% $CO_2$.

## Western blot

Cells were lysed in RIPA buffer (150 mm NaCl, 1.0% NP-40, 50 mm Tris-HCl pH 8.0, 1% sodium dodecyl sulfate, 0.5% sodium deoxycholate) supplemented with protease inhibitor cocktail (Roche) and phosphatase inhibitor cocktail (Roche). Subsequently, SDS loading buffer was added, followed by boiling for 10 min. Antibodies used included p-PKA Substrate, CREB, p-CREB, FLAG tag (Reagents and tools table).

## Immunohistochemistry

Mouse kidneys were fixed in 10% formalin overnight and then embedded in paraffin. Tissue sections were blocked with 3% $H_2O_2$ for 15 min and then blocked by 5% BSA for 1 h at room temperature. Subsequently, the sections were incubated overnight at 4 °C with the following primary antibodies: 8-OHdG, PEG, AQP2, SYNPO, or LRP2 (Reagents and tools table). Following the primary antibody incubation, the sections were incubated with an anti-mouse/rabbit HRP secondary antibody. Finally, the sections were imaged using a microscope (BX51, Olympus).

## DHE staining

For WT and ADPKD mice, adult mouse kidneys were dissected and immediately frozen. The frozen kidneys were then cut into 5–7 μm sections. These sections were treated with a DHE staining solution (Invitrogen, D11347) and incubated for 30 min in a light-protected, humidified box at room temperature. Following the DHE staining, the samples were stained with DAPI and imaged using a fluorescence microscope (DMi8, Leica). For the MDCK cyst model and embryonic kidney cyst model, the culture medium was supplemented with DHE staining solution and incubated for 30 min at 37 °C. The resulting samples were then imaged using the fluorescence microscope (DMi8, Leica).

## Luciferase assay

The CRE-luciferase reporter plasmids were transfected into 293T cells. After a 12-h incubation period, the cell culture medium was supplemented with MMPP and incubated for an additional 24 h. The luciferase activity was then measured using the Dual-Luciferase Reporter Assay System (Promega, E1910).

## Collecting duct epithelial cell isolation

The isolation of renal epithelia cell was performed as described in (Lu et al, 2020). Briefly, mouse kidneys were freshly dissected from WT mice or $Pkd1^{-/-}$ mice, with or without MMPP treatment. The kidneys were minced and digested using the collagenase/hyaluronidase kit (STEMCELL Technologies, 07919). The resulting solution was then treated with biotinylated DBA reagent (Vector Laboratories, B-1035) and incubated at 4 °C for 10 min to facilitate cell binding. Subsequently, the CELLection Biotin Binder Kit (Invitrogen, 11533D) was used to enrich and isolate the DBA-positive renal epithelial cells from the solution.

## Protein purification and pull-down assay

Cherry-CREB, Cherry-CREBΔbZIP, FLAG-CREB, FLAG-CREB Q1, FLAG-CREB KID, FLAG-CREB Q2, FLAG-CREB bZIP, FLAG-CREBΔbZIP proteins were expressed and purified from *E. coli* BL21 (DE3) cells. Briefly, pGEX expression plasmids containing GST tag of the indicated proteins were transformed into BL21 cells. After overnight induction with IPTG at 16 °C, the cells were sonicated in GST lysis buffer [50 mM tris-HCl (pH 8.0), 100 mM NaCl and 1% Triton X-100, 0.2 mM PMSF, and protease inhibitor cocktail]. The cell lysate was centrifuged at 12,000 × g for 20 min at 4 °C. The supernatants were collected and incubated with

Glutathione-Sepharose 4B beads (GE Healthcare, 17-0756-05) at 4 °C overnight. Afterward, the beads were washed three times with lysis buffer. The GST-fusion proteins were then eluted twice using elution buffer containing 10 mM reduced glutathione and 1 mM DTT. To conduct the pull-down assay, the GST tag was cleaved from the fusion proteins. Bio-MMPP was then added to the designated solutions containing FLAG-tagged proteins and incubated at 4 °C overnight. The MMPP-protein complexes were subsequently purified using Streptavidin MagPoly beads (Smart-Lifesciences, SM017010). Finally, the beads were washed and subjected to boiling for Western blotting analysis.

## Streptavidin pull-down assay

Bio-MMPP was added to the culture medium of HEK293T cells and incubated for 24 h. After removing the medium, the cells were washed three times with PBS. Cells were then lysed with NP-40 lysis buffer (containing 150 mM NaCl, 1.0% NP-40, 50 mM Tris-HCl pH 8.0, and a protease inhibitor cocktail) and incubated on ice for 30 min. The lysate was centrifuged at 12,000 × g for 20 min at 4 °C. Streptavidin MagPoly beads (Smart-Lifesciences, SM017010) were added to the supernatant and incubated at 4 °C overnight, followed by washing and boiling for western blotting analysis.

## EMSA assay

The EMSA assay was conducted as described in (Li et al, 2018). In brief, the Cy3-labeled double-stranded CRE DNA probes (5'-AGAGATTGCCTGACGTCAGAGAGCTAG-3') was purchased from Tsingke Biotech. The reaction mixture, composed of 2 nM CRE probes, 2 μg CREB protein, and the indicated amounts of MMPP, was incubated at room temperature in 1× binding buffer (25 mM Tris, pH 7.5, 200 mM NaCl, 5 mM $MgCl_2$, 1 mM DTT, 5% glycerol, and 0.05% Triton X-100) for 20 min. Loading buffer was then added to the reaction mixture, and it was resolved in a 4% native acrylamide/Bis gel in 0.5× TBE buffer (44.5 mM Tris, 44.5 mM boric acid, and 0.5 mM EDTA, pH 8.3). The signals were detected using a chemiluminescence imager (Alliance Q9 advanced, UVITEC).

## In vitro colocalization

The Recombinant Cherry or Cherry-CREB, Cherry-CREBΔbZIP proteins were purified and diluted to a concentration of 40 μM in a buffer containing 20 mM Tris-HCl, 1 mM DTT, and 150 mM NaCl. 10% PEG8000 was added to the recombinant protein solutions, followed by the addition of 100 μg/mL FITC-MMPP. The protein-nanoparticle solutions were then loaded onto a glass slide, covered with a coverslip, and imaged using a fluorescence microscope (LSM900, ZEISS).

## RNA isolation and RT-PCR

Total RNA was extracted from primary renal epithelia cells, 293T cells, or PN cells using TRIzol (Invitrogen, 15596018). Subsequently, 2 μg of the extracted RNA was reverse-transcribed to complementary DN A using the cDNA Synthesis Kit (Roche, 5081955001). Finally, real-time quantitative polymerase chain reaction (RT-PCR) was conducted using SYBR Green Master

(Roche, 41472600). Gene-specific primers are provided in Appendix Table S1.

## RNA-seq analysis

Total RNA was extracted from renal epithelia cells obtained from ADPKD mice, with or without MMPP treatment, as well as from WT mice. RNA sequencing (RNA-seq) was conducted on the Illumina NovaSeq platform. The RNA-seq reads were subsequently aligned to the mouse reference genome (GRCm38/mm10) using HISAT2 (v2.1.0). FeatureCounts v1.6.0 was used to count the mapped reads. DESeq2 was used to calculate the differentially expressed genes, which were considered significant if they had a fold change of $\leq -1.5$ or $\geq 1.5$ and $P < 0.05$. GO enrichment analysis was performed on DAVID.

## ChIP-qPCR

Primary renal epithelia cells, HEK293T cells or WT 9-12 cells were fixed with 1% formaldehyde at room temperature for 10 min. Crosslinking was stopped by adding 150 mM glycine to the solution at room temperature for 5 min. The cells were then washed twice with PBS and collected using ChIP lysis buffer, which contained 1% SDS, 10 mM EDTA, 50 mM Tris-HCl pH 8.0, and 1% protease inhibitor cocktail. The cell solution was sonicated using a Bioruptor Sonicator to fragment the DNA to 200–500 bp. Immunoprecipitation was performed using 2 μg of p-CREB antibody. After washing, the Dynabeads were eluted using elution buffer, which contained 1% sodium dodecyl sulfate and 0.1 M NaHCO$_3$. The cross-linking was then reversed by incubating the eluted material at 65 °C overnight. DNA was extracted for further RT-PCR analysis after elution and reversed cross-linking. Gene-specific primers are provided in Appendix Table S2.

## 3D MDCK cyst model

MDCK I cells were seeded at a density of 400 cells/well in 400 μL medium containing 3 mg/mL Type I collagen (Sigma-Aldrich, C2124), 10 mM HEPEs, and 27 mM NaHCO$_3$ in 24-well plates. The plates were incubated at 37 °C for 90 min to allow gel formation. Then, 1 mL of DMEM/F12 culture medium supplemented with 10% FBS and 10 μM forskolin (Sigma-Aldrich, F3917) was added to each well of the plates. The cells were cultured for 12 days with the medium changed daily. The indicated concentration of MMPP was added to the culture medium from day 4 to day 12. Micrographs were acquired at day 4, 6, 8, 10, and 12 using a microscope (DMi8, Leica).

## Embryonic kidney cyst model

The embryonic kidney cysts were induced following the protocol described in (Sun et al, 2019b). Briefly, kidneys were dissected from CD1 mouse embryos at E13.5 and seeded into 0.4 μm transwell inserts (Corning, 3401) placed in 12-well plates. To each well, 400 μL of DMEM/F12 culture medium was added, which was supplemented with 10 mM HEPEs (Sigma-Aldrich, H3375), 2 mM L-glutamine (Sigma-Aldrich, G7513), 1/100 insulin-transferrin-selenium (Life, 41400-045), 1/40 penicillin/streptomycin (Life, 15140-122), 25 ng/ml prostaglandin E1 (Sigma-Aldrich, P5515), 32 pg/ml T3 (Sigma-Aldrich, T5516), and 100 μM 8-Br-cAMP

(Sigma-Aldrich, B5386). The indicated concentration of MMPP was then added to the culture medium. The medium was changed twice a day, and the kidneys were cultured for 6 days in a 37 °C incubator with 5% CO$_2$. On days 0, 2, 4, and 6, images were acquired using a microscope (XI71, Olympus). The cystic index (cyst area/total area) was determined by measuring and calculating based on the images using ImageJ software.

## Mice and treatment

$Pkd1^{fl/fl}$;$Cre/Esr1^{+}$ mice were generated as described previously in (Yang et al, 2018). Mice were maintained under a standard 12-h light/dark cycle with free access to food and water. Tamoxifen (Sigma-Aldrich, T5648) was dissolved in corn oil and administered to the male mice via intraperitoneal (IP) injection at a dosage of 250 mg/kg at P25 and P28. This treatment was done to activate Cre recombinase and delete the $Pkd1$ gene. Two genotypes of mice ($Pkd1^{+/+}$ and $Pkd1^{-/-}$) were each randomly divided into two groups. For the administration of MMPP, MMPP was dissolved in 100 μL of saline solution and intravenously (IV) injected into ADPKD mice at a dosage of 10 mg/kg. The injections were given every 4 days, starting from day 55 and continued for a period of 2 months.

On day 115, the mice were sacrificed, and blood, liver, and kidney samples were collected for further analysis. The cyst area and total area were measured by visualizing hematoxylin and eosin-stained kidney and liver sections, and the measurements were evaluated using ImageJ software. Blood urea nitrogen (BUN) levels were measured using the Urea Assay Kit (Nanjing Jiancheng Bioengineering, C013-1-1). Outliers were detected by identifying those with a z-score exceeding 3 utilizing the outlier test, no mice were excluded in this study. During the animal experiment, four individuals were involved. Y.S. was responsible for randomly assigning 18 mice to both the control group and the experimental group, respectively, as well as for drug preparation. Q.Z. independently handled drug injections using a 32 G needle to minimize mouse discomfort and mouse cages were placed randomly. Due to significant color differences between Saline and MMPP, the experimenter could not be blinded to whether the mouse was injected with Saline or MMPP. Y.Y. independently collected animal tissues, while X.D. conducted data analysis independently. All mouse studies were approved by the Ethical Committee of Tianjin Medical University (permit number SYXK: 2020-0010).

## Statistical analysis

Statistical analysis was performed using SPSS software. The data were presented as means ± SD from a minimum of three independent experiments. For the comparison of two sets of data, if they conform to a normal distribution, a two-tailed unpaired Student's $t$ test was employed; if not, a non-parametric test with Mann-Whitney U test was applied. In the case of a single-factor multiple-group data comparison, one-way ANOVA was utilized when the data follows a normal distribution (with homogeneity of variance tested using LSD test for equal variances and Dunnett's T3 test for unequal variances), and a non-parametric test was applied when normal distribution assumptions were not met. For the comparison of multiple groups with two factors, two-way ANOVA analysis was conducted. In the context of repeated measurements, a repeated measures ANOVA analysis was employed. A $P$-value of less than 0.05 was considered statistically significant.

## The paper explained

### Problem

Autosomal Dominant Polycystic Kidney Disease (ADPKD) is the most prevalent genetic kidney disorder globally, affecting approximately 12.5 million people. About 50% of individuals diagnosed with ADPKD are likely to progress to end-stage renal disease by their 50s or 60s. ADPKD involves multiple complex pathways, and treatments that target only one pathway often fail to effectively halt disease progression. Therefore, a significant challenge in the field is to develop safe, long-term treatments that can simultaneously target several key pathological pathways involved in ADPKD.

### Results

In this study, we demonstrated that ultra-small polyethylene glycol-incorporated $Mn^{2+}$-chelated melanin-like nanoparticles (MMPP) effectively inhibit cyst growth in an orthologous ADPKD mouse model. Upon systemic injection, MMPP crosses the glomerular filtration barrier and specifically accumulates in renal tubules. Beyond its classical anti-oxidant properties, we identified the cAMP-response element-binding protein (CREB) as a key intracellular target of MMPP. By directly binding to the bZIP domain of CREB, MMPP inhibits its transcriptional activity and the expression of downstream genes. Thus, MMPP targets both oxidative stress and CREB, two key pathways implicated in cystogenesis, leading to effective treatment of ADPKD in mice.

### Impact

Our study enhances the therapeutic potential of melanin-like nano-particles (MNPs) by uncovering specific intracellular targets and identifying new indications, such as ADPKD. This expands their applicability beyond chronic diseases associated with oxidative stress, to conditions involving dysregulated cAMP-CREB signaling. Our findings deepen our understanding of MNPs' mechanisms of action, paving the way for the development of safer and more effective melanin-based nanomedicines.

## Data availability

RNA sequencing data were deposited in the Gene Expression Omnibus platform with accession number GSE235269. All other data generated or analyzed during this study are included in this published article (and its supplementary information files).

The source data of this paper are collected in the following database record: biostudies:S-SCDT-10_1038-S44321-024-00167-2.

## Peer review information

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

## Acknowledgements

The authors thank the core facility of the Department of Laboratory Animal Science for their technical support with animal studies. This work was supported by grants from the National Natural Science Foundation of China (32070634 to LZ, 82071982 to SS, 92068113, 32241018, and 82070689 to YC, 82200832 to YS, and 82200834 to ZL), the Natural Science Foundation of Tianjin (21JCJQJC00100 to LZ, 19JCJQJC63700 to SS, and 19JCJQJC63800 to YC), Tianjin Municipal Education Commission (2022ZD054 to LZ) and China Postdoctoral Science Foundation (2021M692407 to ZL).

## Author contributions

**Lirong Zhang**: Conceptualization; Supervision; Funding acquisition; Writing—original draft; Project administration; Writing—review and editing. **Yongzhan Sun**: Resources; Data curation; Formal analysis; Funding acquisition; Investigation; Methodology; Writing—original draft. **Quan Zou**: Resources; Data curation; Formal analysis; Validation; Investigation; Methodology. **Huizheng Yu**: Data curation; Software; Investigation. **Xiaoping Yi**: Data curation; Investigation. **Xudan Dou**: Software; Validation. **Yu Yang**: Data curation; Investigation; Methodology. **Zhiheng Liu**: Funding acquisition; Methodology. **Hong Yang**: Methodology. **Junya Jia**: Resources. **Yupeng Chen**: Conceptualization; Funding acquisition; Visualization; Writing—original draft; Project administration; Writing—review and editing. **Shao-Kai Sun**: Conceptualization; Funding acquisition; Writing—original draft; Project administration; Writing—review and editing. **Lirong Zhang**: Conceptualization; Supervision; Funding acquisition; Writing—original draft; Project administration; Writing—review and editing.

Source data underlying figure panels in this paper may have individual authorship assigned. Where available, figure panel/source data authorship is listed in the following database record: biostudies:S-SCDT-10_1038-S44321-024-00167-2.

## Disclosure and competing interests statement

The authors declare no competing interests.

# Expanded View Figures

**Figure EV1.  MMPP inhibits cyst growth and scavenges ROS in in vitro and ex vivo ADPKD models.**

(A) Representative images (left) and cyst diameter (right) of MDCK cysts treated with the indicated doses of MMPP (representative of eight independent experiments; $n = 8$ biological replicates). (B) DHE staining (left) and quantification (right) of cysts in each of the indicated groups (representative of five independent experiments; $n = 5$ biological replicates). (C) Representative images (left) of mouse embryonic kidneys treated with the indicated doses of MMPP and quantification (right) of the percentage of cyst area relative to total kidney area (representative of three independent experiments; $n = 3$ biological replicates). (D) DHE staining (left) and quantification (right) of embryonic kidneys on day 6 of MMPP treatment (representative of three independent experiments; $n = 3$ biological replicates). Scale bars, 100 μm (A and B) and 1 mm (C and D). Data presented as means ± SD. One-way ANOVA with LDS test or Dunnett's T3 test was used for statistical analysis in (A), (B) and (D), repeated measures ANOVA with LSD test was used for statistical analysis in (C).

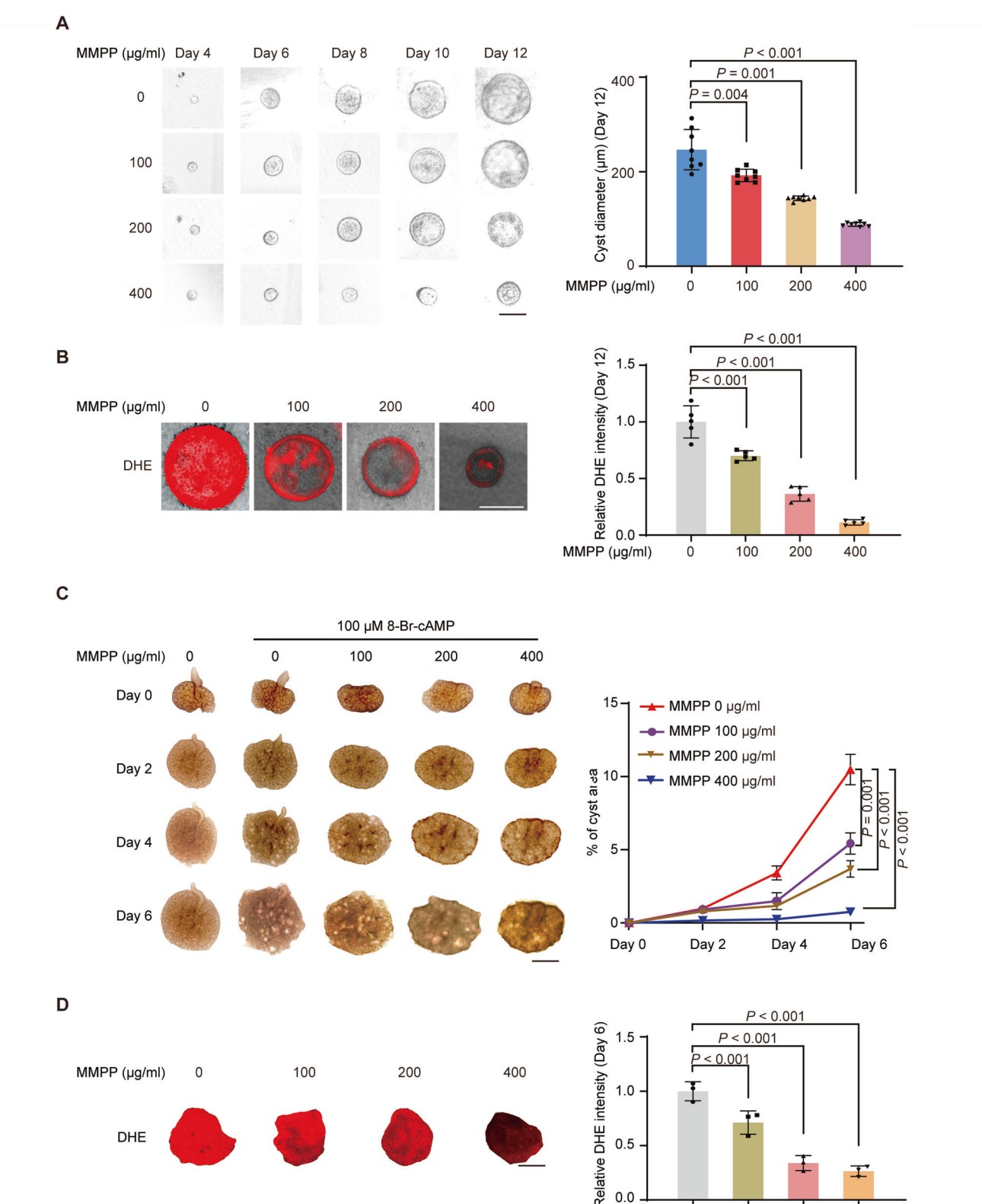

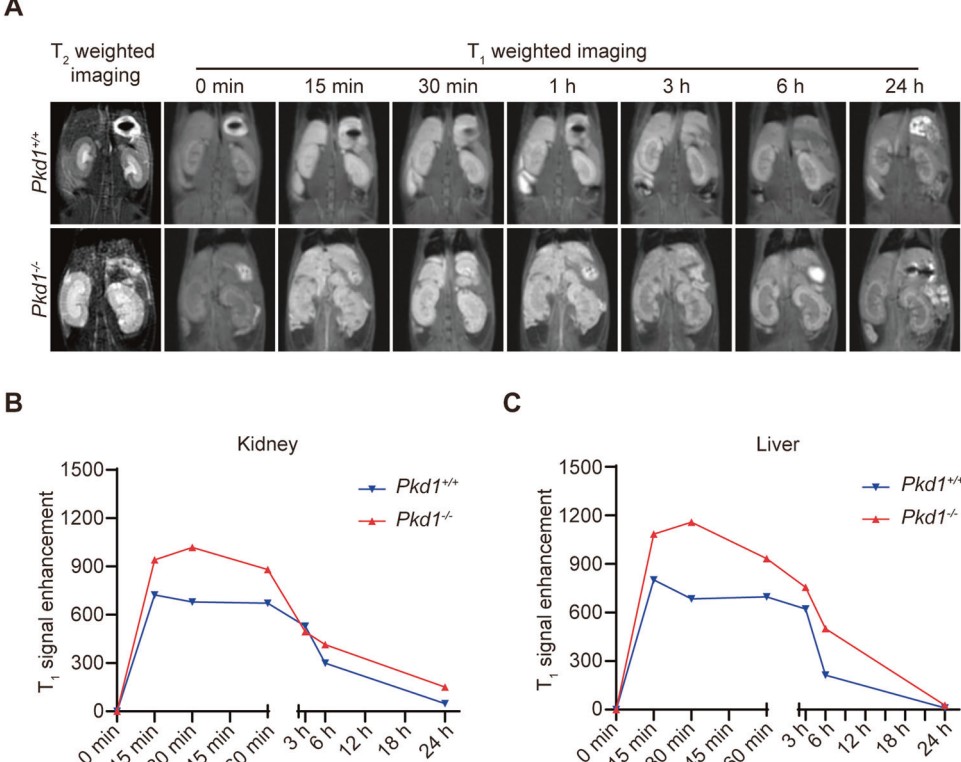

**Figure EV2. MR imaging of MMPP distribution in kidneys and liver.**

(**A**) MR images of MMPP in *Pkd1*$^{+/+}$ and *Pkd1*$^{-/-}$ mice (Dosage: 100 mg/kg). (**B**) Quantification of MMPP T$_1$ signal in kidneys. (**C**) Quantification of MMPP T$_1$ signal in livers.

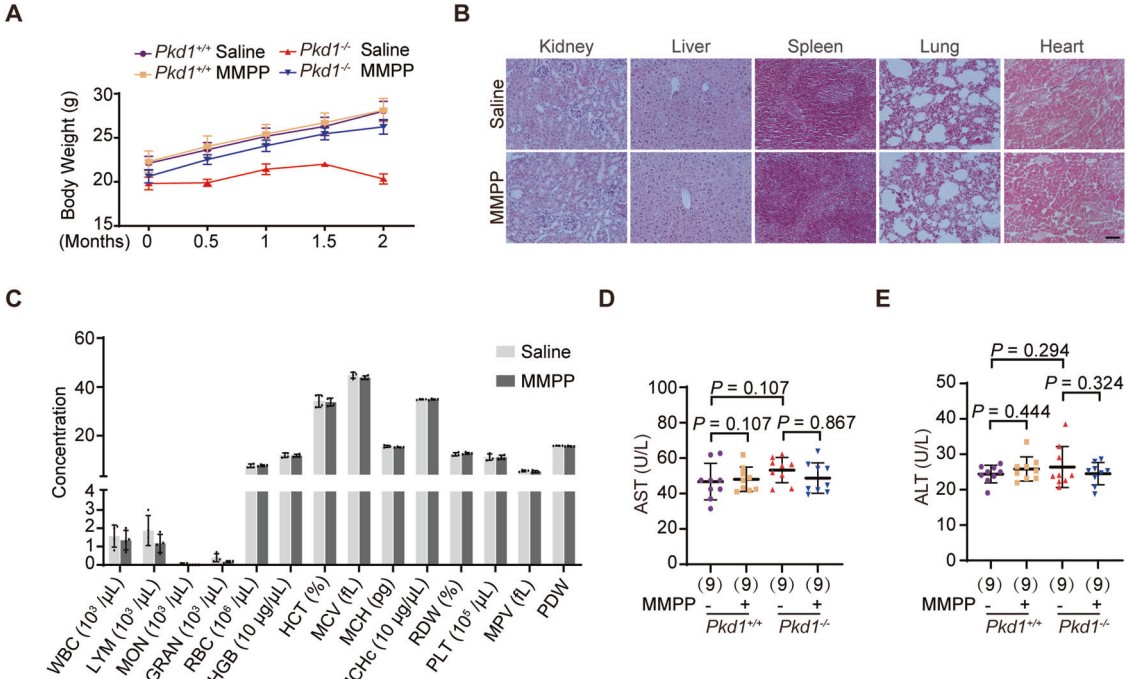

**Figure EV3. Safety assessment of MMPP in vivo.**

(A) Growth curve of mouse body weight of *Pkd1*$^{+/+}$ and *Pkd1*$^{−/−}$ mice treated with saline or MMPP ($n \geq 3$) (B) H&E staining of organs from mice treated with Saline or MMPP. (C) Hematology analysis of whole blood in MMPP-treated mice ($n = 4$). (D, E) AST (D) and ALT (E) levels in MMPP-treated mice from the indicated groups ($n = 9$). Scale bar, 50 μm. Data presented as means ± SD. Two-way ANOVA with LSD test was used for statistical analysis.

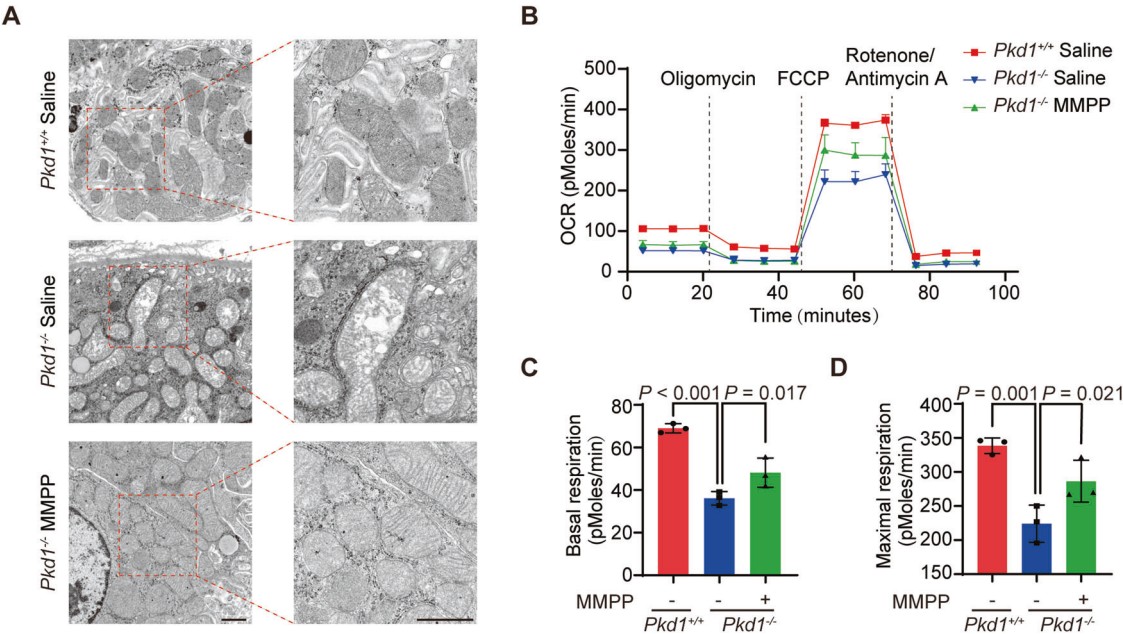

**Figure EV4.  MMPP treatment improves mitochondria morphology and metabolic function in ADPKD kidneys.**

(**A**) TEM images of mitochondria from kidney tissues of the indicated groups. (**B**) Measurement of the mitochondrial OCR of renal primary tubule cells isolated from the indicated groups (representative of three independent experiments; $n = 3$ biological replicates). (**C**) Basal respiration of mitochondria from the indicated groups (representative of three independent experiments; $n = 3$ biological replicates). (**D**) Maximal respiration of mitochondria from the indicated groups (representative of three independent experiments; $n = 3$ biological replicates). Scale bars, 1 µm. Data presented as means ± SD. One-way ANOVA with LSD test was used for statistical analysis in (**C**) and (**D**).

