## [Peer Review File · EMBO Molecular Medicine]

Melanin-like nanoparticles slow cyst growth in ADPKD by dual inhibition of oxidative stress and CREB

Lirong Zhang, Yongzhan Sun, Quan Zou, Huizheng Yu, Xiaoping Yi, Xudan Dou, Yu Yang, Zhiheng Liu, Hong Yang, Junya Jia, Yupeng Chen, and Shao-kai Sun

Corresponding authors: Lirong Zhang (lzhang@tmu.edu.cn) , Yupeng Chen (ychen@tmu.edu.cn), Shao-kai Sun (shaokaisun@tmu.edu.cn)

Review Timeline:

Submission Date:	12th Jul 24
Editorial Decision:	5th Aug 24
Revision Received:	14th Sep 24
Editorial Decision:	9th Oct 24
Revision Received:	18th Oct 24
Accepted:	18th Oct 24

Editor: Lise Roth

Transaction Report:

5th Aug 2024

Dear Dr. Zhang,

Thank you for the submission of your manuscript to EMBO Molecular Medicine. We have now received feedback from the three reviewers who agreed to evaluate your manuscript. As you will see from the reports below, the referees acknowledge the interest of the study and are overall supporting publication of your work pending appropriate revisions.

Addressing the reviewers' concerns in full will be necessary for further considering the manuscript in our journal. However, following further cross-commenting with the referees, we agreed that comparison with other antioxidant treatments in ADPKD was beyond the scope of your study, and addressing this concern from referee #3 will therefore NOT be requested for further consideration of the manuscript. Acceptance of the manuscript will entail a second round of review. EMBO Molecular Medicine encourages a single round of revision only and therefore, acceptance or rejection of the manuscript will depend on the completeness of your responses included in the next, final version of the manuscript. For this reason, and to save you from any frustrations in the end, I would strongly advise against returning an incomplete revision.

We are expecting your revised manuscript within three months, if you anticipate any delay, please contact us.

We require:

4) A .docx formatted letter INCLUDING the reviewers' reports and your detailed point-by-point responses to their comments. As part of the EMBO Press transparent editorial process, the point-by-point response is part of the Review Process File (RPF), which will be published alongside your paper.

5) A complete author checklist, which you can download from our author guidelines (<https://www.embopress.org/page/journal/17574684/authorguide#submissionofrevisions>). Please insert information in the checklist that is also reflected in the manuscript. The completed author checklist will also be part of the RPF.

6) All Materials and Methods need to be described in the main text using our 'Structured Methods' format, which is required for all research articles. According to this format, the Methods section includes a Reagents and Tools Table (listing key reagents, experimental models, software and relevant equipment and including their sources and relevant identifiers) followed by a Methods and Protocols section describing the methods using a step-by-step protocol format. The aim is to facilitate adoption of the methodologies across labs. More information on how to adhere to this format as well as a downloadable template (.docx) for the Reagents and Tools Table can be found in our author guidelines:
<https://www.embopress.org/page/journal/17574684/authorguide#structuredmethods>

7) Please note that all corresponding authors are required to supply an ORCID ID for their name upon submission of a revised manuscript.

8) It is mandatory to include a 'Data Availability' section after the Materials and Methods. Before submitting your revision, primary datasets produced in this study need to be deposited in an appropriate public database, and the accession numbers and database listed under 'Data Availability'. Please remember to provide a reviewer password if the datasets are not yet public (see <https://www.embopress.org/page/journal/17574684/authorguide#dataavailability>).

In case you have no data that requires deposition in a public database, please state so in this section. Note that the Data

Availability Section is restricted to new primary data that are part of this study.

9) For data quantification: please specify the name of the statistical test used to generate error bars and P values, the number (n) of independent experiments (specify technical or biological replicates) underlying each data point and the test used to calculate p-values in each figure legend. The figure legends should contain a basic description of n, P and the test applied. Graphs must include a description of the bars and the error bars (s.d., s.e.m.). Please provide exact p values.

10) Our journal encourages inclusion of *data citations in the reference list* to directly cite datasets that were re-used and obtained from public databases. Data citations in the article text are distinct from normal bibliographical citations and should directly link to the database records from which the data can be accessed. In the main text, data citations are formatted as follows: "Data ref: Smith et al, 2001" or "Data ref: NCBI Sequence Read Archive PRJNA342805, 2017". In the Reference list, data citations must be labeled with "[DATASET]". A data reference must provide the database name, accession number/identifiers and a resolvable link to the landing page from which the data can be accessed at the end of the reference. Further instructions are available at .

11) We replaced Supplementary Information with Expanded View (EV) Figures and Tables that are collapsible/expandable online. A maximum of 5 EV Figures can be typeset. EV Figures should be cited as 'Figure EV1, Figure EV2' etc... in the text and their respective legends should be included in the main text after the legends of regular figures.

12) The paper explained: EMBO Molecular Medicine articles are accompanied by a summary of the articles to emphasize the major findings in the paper and their medical implications for the non-specialist reader. Please provide a draft summary of your article highlighting

13) Author contributions: CRedit has replaced the traditional author contributions section because it offers a systematic machine readable author contributions format that allows for more effective research assessment. Please remove the Authors Contributions from the manuscript and use the free text boxes beneath each contributing author's name in our system to add specific details on the author's contribution. More information is available in our guide to authors.

16) As part of the EMBO Publications transparent editorial process initiative (see our Editorial at <http://embomolmed.embopress.org/content/2/9/329>), EMBO Molecular Medicine will publish online a Review Process File (RPF) to accompany accepted manuscripts.

In the event of acceptance, this file will be published in conjunction with your paper and will include the anonymous referee reports, your point-by-point response and all pertinent correspondence relating to the manuscript. Let us know whether you agree with the publication of the RPF and as here, if you want to remove or not any figures from it prior to publication. Please note that the Authors checklist will be published at the end of the RPF.

EMBO Molecular Medicine has a "scooping protection" policy, whereby similar findings that are published by others during review or revision are not a criterion for rejection. Should you decide to submit a revised version, I do ask that you get in touch

after three months if you have not completed it, to update us on the status.

I look forward to receiving your revised manuscript.

Yours sincerely,

Lise Roth

***** Reviewer's comments *****

Referee #1 (Comments on Novelty/Model System for Author):

The studies in the mouse are clearly performed in the adequate model system. The in vitro and ex vivo data are not that compelling in the view of this reviewer given that they do not reflect the PKD manifestation.

Referee #1 (Remarks for Author):

In this paper by Sun et al, the authors describe the use of nanoparticles carrying melanin as a novel therapeutic approach to be used for polycystic kidney disease.

The study is overall interesting as it describes a repurposing of this approach, pioneered in the cancer setting, for a different disease. The study is based on the biodistribution of novel melanin-like nanoparticles into the kidney (both cortex and medulla) and in the liver. The authors report that the approach is effective in retarding disease progression in an inducible PKD mouse model and they further show that, at least part of the activity, is mediated by reducing ROS activity and inhibiting CREB activity. The therapeutic approach is novel and the efficacy well documented. There are some important shortcomings in the second part of the study where the authors attempt at demonstrating a direct interaction between the nanoparticles and the transcription factor CREB as a possible mechanism of action of the complexes.

Major:

- Figure 1 shows the biodistribution of the MMP particles, while Figure 2 shows an effect in retarding growth in a model of cyst expansion driven by forskolin in MDCK cells and in an organoid culture where cysts are induced by forskolin, both in the context of wild-type pkd1. These readouts are of limited significance, especially when considering that in subsequent experiments the authors use the much more reliable in vivo model of induced Pkd1 inactivation.
- Figure 3 is very interesting and well conducted, investigators use an inducible mouse model of PKD and treat the mice for 3 months, every 4 days, showing efficacy of MMP in retarding disease progression in PKD. Figure 3D is it showing the average situation? Since there is a variability both in the K/BW and in the cystic index, it would be interesting and important to show which of the animals (i.e. which dot in figure 3E) is the shown specimen. Similar considerations apply to figure 3H.
- Figure 4 is very interesting and one of the central parts of the study. The authors show that the genes in cluster 2, i.e. downregulated in PKD and restored by MMP treatment are enriched in genes important for mitochondrial activity in line with previous work. About 23% of genes in that cluster belong to mitochondrial dysfunction. What else is present in the remaining 77%? And could that be relevant for the rescue as well? Also, there are a large number of genes that were not changing in PKD kidneys, but are heavily deregulated upon MMP administration. What is in there? Is there anything popping up that might represent a concern for toxicity?
- Similar reasoning as above on the cluster 4 data shown in figure 5. 31% of genes are direct targets of CREB. This is extremely interesting and also in view of the mechanism proposed in the subsequent figures it is somewhat surprising that MMP does not result in toxicity if it has such a strong effect on CREB, essential component in the response to multiple GPCRs. Changes shown in Figure 5C are quite impressive as most if not all genes found deregulated in PKD are reversed. Perhaps an HCA showing all changes, also the ones not restored would provide an important comparison. In general related to the transcriptional studies reported in Figure 4 as well, a general HCA would give a sense of the overall changes observed.

- Data in figure 6 and 7 are not well controlled. In figure 6F and 7C, what happens if the same input is added to non-biotinylated MMP, followed by streptavidin pull-down. How is the background of a non-specific pull-down? Ideally, one would want to have a defective MMP as a negative control.

- Finally, a conceptual point. In Figure 1 the authors show a distribution of MMP that is both in the cortex and in the medulla. The accumulation is much more robust in the cortex than in the medulla, and in both cases it is cytosolic. If the interaction occurs directly with CREB, however, the MMP must be reaching the nucleus. And indeed the scheme in figure 8 shows a nuclear localization. Do the authors see an accumulation in the nucleus? How would stoichiometrically be explained the inhibition of CREB, particularly relevant in the medulla were the cAMP effect is believed to be maximal in PKD?

Referee #2 (Comments on Novelty/Model System for Author):

Sun et al report that that ultra small polyethylene glycol incorporated Mn²⁺ chelated MNP (MMPP) traverse the glomerular filtration barrier and accumulate in renal tubules. They inhibited in vitro, ex vivo cystogenesis, and in vivo cystogenesis by a dual effect of melanin, as a highly effective antioxidant and by directly binding to the bZIP DNA binding domain of CREB, leading to competitive inhibition of CREB's DNA binding ability and subsequent reduction in CREB target gene expression. In addition, they engineered an orally administrable formulation that was as effective as the intraperitoneal administration. The studies were well conducted. Many observations are novel and, if confirmed, likely to have a substantial impact on the field.

Referee #3 (Remarks for Author):

In this manuscript, Sun et al. demonstrate the utilization of melanin-like nanoparticles (MNPs) for treating ADPKD. The authors have synthesized an ultrasmall MNP (MMPP) that efficiently crosses the glomerular filtration barrier and accumulates in the renal tubules. Their findings suggest that MMPP can significantly delay the progression of ADPKD by reducing oxidative stress and inhibiting the cAMP-CREB pathway. The manuscript proposes a novel therapeutic approach for ADPKD and identifies new intracellular targets for MNPs. The manuscript could be further improved if the authors address the following aspects:

1. The manuscript's primary finding is that MMPP inhibits CREB transcriptional activity, in addition to its antioxidant effects. To reinforce this conclusion, the authors should consider conducting control experiments using established antioxidants for ADPKD treatment. This would more convincingly demonstrate the specific impact of MMPP on CREB inhibition.

2. Increased oxidative stress and activation of the cAMP signaling pathway are also significant contributors to polycystic liver disease (PLD). However, since liver cysts originate from cholangiocytes, it is essential to investigate whether MMPP accumulates in cholangiocytes to validate its efficacy in treating PLD.

3. During the progression of ADPKD, tubular injury can further exacerbate the disease. Therefore, the authors need to evaluate the impact of MMPP treatment on kidney injury.

4. MNPs originate from various sources and thus exhibit diverse biological effects. Could the type of melanin used in the preparation of MMPP influence the results?

5. The authors should include the genes from clusters 1-6 in the supplementary information to provide comprehensive data access.

6. While oxidative stress is a key factor in PKD pathogenesis, antioxidant therapies have failed in ADPKD clinical trials. The authors should compare previous studies in the discussion section, elucidating how MMPP differs from previous studies and why it might be better suited for clinical translation.

Referee #1 (Comments on Novelty/Model System for Author):

The studies in the mouse are clearly performed in the adequate model system. The in vitro and ex vivo data are not that compelling in the view of this reviewer given that they do not reflect the PKD manifestation.

Response: In this study, we employed an *in vitro* 3D MDCK model and an *ex vivo* embryonic kidney model to demonstrate that MMPP treatment effectively reduces intracellular ROS levels. While these models are widely used in the field and provide valuable mechanistic insights, we agree with the reviewer that they do not fully capture the manifestation of PKD. We acknowledge that the *in vivo* results offer a more accurate and meaningful reflection of the disease. Therefore, we have moved the *in vitro* and *ex vivo* data to the Expanded View Figure section to support the overall findings without overemphasizing these models.

Referee #1 (Remarks for Author):

In this paper by Sun et al, the authors describe the use of nanoparticles carrying melanin as a novel therapeutic approach to be used for polycystic kidney disease.

The study is overall interesting as it describes a repurposing of this approach, pioneered in the cancer setting, for a different disease. The study is based on the biodistribution of novel melanin-like nanoparticles into the kidney (both cortex and medulla) and in the liver. The authors report that the approach is effective in retarding disease progression in an inducible PKD mouse model and they further show that, at least part of the activity, is mediated by reducing ROS activity and inhibiting CREB activity. The therapeutic approach is novel and the efficacy well documented. There are some important shortcomings in the second part of the study where the authors attempt at demonstrating a direct interaction between the nanoparticles and the transcription factor CREB as a possible mechanism of action of the complexes.

Major:

*- Figure 1 shows the biodistribution of the MMP particles, while Figure 2 shows an effect in retarding growth in a model of cyst expansion driven by forskolin in MDCK cells and in an organoid culture where cysts are induced by forskolin, both in the context of wild-type *pkd1*. These readouts are of limited significance, especially when considering that in subsequent experiments the authors use the much more reliable *in vivo* model of induced *Pkd1* inactivation.*

Response: As mentioned in our response to the initial comment, we have moved the *in vitro* and *ex vivo* data to the Expanded View Figure section to provide supplementary support for the overall findings. While these models offer some mechanistic insights, we agree that the *in vivo* model of inducible *Pkd1* inactivation provides a more reliable and clinically relevant system for assessing the therapeutic potential of MMPP. The *in vivo* data are the primary focus of this study, as they more accurately reflect the disease context and progression.

- Figure 3 is very interesting and well conducted, investigators use an inducible mouse model of PKD and treat the mice for 3 months, every 4 days, showing efficacy of MMP in retarding disease progression in PKD. Figure 3D is it showing the average situation? Since there is a variability both in the K/BW and in the cystic index, it would be interesting and important to show which of the animals (i.e. which dot in figure3E) is the shown specimen. Similar considerations apply to figure 3H.

Response: Thank you for your detailed feedback on Figure 3 (now Figure 2 in the revised manuscript) in our original manuscript. We appreciate the opportunity to clarify and enhance our presentation of these results.

Regarding your observation of the images in panels D and H, we agree that it is crucial to explicitly indicate which data points these images represent from graphs E and I. Panel D is showing nearly the average situation. We specify this correspondence in the figure and figure legend to ensure clarity and transparency about the sample selection (The representative

images are highlighted in black in C, E and I).

- Figure 4 is very interesting and one of the central parts of the study. The authors show that the genes in cluster 2, i.e. downregulated in PKD and restored by MMP treatment are enriched in genes important for mitochondrial activity in line with previous work. About 23% of genes in that cluster belong to mitochondrial dysfunction. What else is present in the remaining 77%? And could that be relevant for the rescue as well? Also, there are a large number of genes that were not changing in PKD kidneys, but are heavily deregulated upon MMP administration. What is in there? Is there anything popping up that might represent a concern for toxicity?

Response: Thank you for your valuable suggestion. We performed Gene Ontology (GO) term enrichment analysis on the remaining 67.7% of genes located outside the mitochondria in cluster 2. As shown in the following Figure 1A, these genes were significantly enriched in pathways related to cilium organization, cellular catabolic process, RNA splicing and fatty acid metabolic process. Many of these pathways are associated with PKD pathology (Ming Ma., *Semin Cell Dev Biol.*, 2021; Haumann et al., *Int J Mol Sci.*, 2020; Gonzalez-Paredes et al., *Gene*, 2014; Menezes et al., *EBioMedicine*, 2016), and are located in various cellular compartments, including the nucleus, cytoplasm, and other organelles. The restoration of these pathways could contribute positively to the rescue effect in ADPKD.

For the genes in Cluster 1, which showed minimal changes in ADPKD but were significantly downregulated following MMPP treatment, GO term enrichment analysis (Figure 1B) revealed that they are involved in pathways such as angiogenesis, extracellular matrix deposition, Wnt/Notch/Erk signaling, NF-kappa B signaling, and oxidative stress (Huang et al., *Pediatr Nephrol.*, 2013; Chea et al., *Yonsei Med J.*, 2009; Ao Li, *ARC Journal of Nephrology.*, 2018; Idowu et al., *Sci. Rep.*, 2018; Yamaguchi et al., *Am J Physiol Renal Physiol.*, 2010; Ta et al., *Nephrology (Carlton)*, 2013; Andries et al., *Pediatr Nephrol.*, 2019). Since these pathways are involved in the progression of PKD, their downregulation may contribute to the therapeutic benefit of MMPP treatment.

For genes in Cluster 5, which exhibited minimal changes in ADPKD but were significantly upregulated following MMPP treatment, we also conducted GO term enrichment analysis. As shown in Figure 1C, many of these genes are involved in pathways related to PKD disease progression such as fatty acid/lipid metabolism, oxidative phosphorylation, recombinational repair and glucose homeostasis (Menezes et al., *EBioMedicine*, 2016; Podrini et al., *Communications Biology*, 2018; Lakhia et al., *Am J Physiol Renal Physiol.*, 2018; Bacolla et al., *J Biol Chem.*, 2001; Rowe et al., *Nat Med.*, 2013). Given that dysregulation of these pathways is associated with PKD progression, the MMPP-induced upregulation of these genes may play a protective role in disease mitigation.

Taken together, MMPP reduced the expression of many cyst-promoting genes while enhancing the expression of protective genes in PKD. Although we have not yet identified any specific toxicity-related pathways in MMPP-treated kidneys, further detailed *in vivo* studies are warranted to fully assess any potential concerns related to toxicity.

Figure 1. MMPP protects the kidney by modulating the expression of disease-associated genes. (A) Gene Ontology (GO) term enrichment analysis of the genes located outside the mitochondria in Cluster 2. (B) GO term enrichment analysis of the genes in Cluster 1. (C) GO term enrichment analysis of the genes in Cluster 5.

- Similar reasoning as above on the cluster 4 data shown in figure 5. 31% of genes are direct

targets of CREB. This is extremely interesting and also in view of the mechanism proposed in the subsequent figures it is somewhat surprising that MMPP does not results in toxicity if it has such a strong effect on CREB, essential component in the response to multiple GPCRs. Changes shown in Figure 5C are quite impressive as most if not all genes found deregulated in PKD are reversed. Perhaps an HCA showing all changes, also the ones not restored would provide an important comparison. In general related to the transcriptional studies reported in Figure 4 as well, a general HCA would give a sense of the overall changes observed.

Response: We appreciate the reviewer's insightful comment regarding the potential toxicity associated with MMPP's strong effect on CREB, a crucial regulator in multiple GPCR signaling pathways. Several factors may explain the lack of observed toxicity in our model. First, due to its size and biodistribution characteristics, MMPP primarily accumulates in the kidney and liver, with minimal off-target effects on other organs. This organ-specific localization limits the potential for systemic toxicity, as its primary impact is focused on the disease-relevant tissues. Additionally, in cystic cells, where cAMP levels are elevated compared to normal cells, CREB phosphorylation and transcriptional activity are abnormally increased. MMPP likely exerts a more pronounced inhibitory effect on this hyperactivated CREB signaling in cystic cells while sparing normal CREB activity in healthy tissues, reducing the potential for side effects. Although we did not observe overt toxicity in our model, we recognize the importance of long-term studies to fully assess any potential side effects, and we have added this consideration to the discussion section.

In the original manuscript, the heatmap in Figure 5C (now Figure 4C in the revised version) shows the expression of all CREB targets specifically in Cluster 4, not across all clusters. These CREB targets make up nearly one-third of Cluster 4 when comparing ADPKD mice to normal mice. Importantly, the expression of these genes was largely restored in ADPKD mice treated with MMPP. In response to your suggestion, we have now performed hierarchical cluster analysis (HCA) on all CREB targets in Cluster 4. As shown in Figure 2A bellow, nearly all of these genes were rescued by MMPP treatment, with the exception of five genes (*Klhl18*, *Klhl26*, *Vps18*, *Dusp5*, and *Hnrnpull1*) at the bottom. These genes are associated with

tumor suppression (Jiang et al., *Cell Biosci*, 2020), ubiquitination and degradation of substrates in cardiomyocytes or cancer (Thareja et al., *Am J Physiol Heart Circ Physiol*, 2023; Senft et al., *Nat Rev Cancer*, 2018), ERK1/2 signaling inhibition (Rushworth et al., *Proc Natl Acad Sci*, 2014), and the ATR signaling pathway in DNA repair (Polo et al., *Mol Cell*, 2013). We have not yet identified a direct link between the expression of these genes and PKD progression. This updated analysis has been included in the revised version.

We fully agree with your suggestion to provide a broader perspective on the transcriptional changes. To this end, we conducted HCA on all the genes across the six clusters. As shown in Figure 2B, the heatmap presents the overall expression changes, with most genes following the patterns observed in their respective line graphs, except for a small subset in Clusters 3 and 6. This updated analysis has been included in the revised version to offer a more comprehensive view of the transcriptional landscape following MMPP treatment.

Figure 2. Heatmap showing the expression of all genes based on hierarchical cluster analysis (HCA). (A) HCA of CREB target genes in Cluster 4. (B) Gene expression and HCA of all of the genes in cystic cells isolated from WT and *Pkd1*^{-/-} mice treated with saline or MMPP.

- Data in figure 6 and 7 are not well controlled. In figure 6F and 7C, what happens if the same input is added to non-biotinylated MMP, followed by streptavidin pull-down. How is the background of a non-specific pull-down? Ideally, one would want to have a defective MMP as

a negative control.

Response: In Figure 6F of the original manuscript (now Figure 5F in the revised manuscript), cells in lanes 1 and 3 were indeed cultured with non-biotinylated MMPP, while those in lanes 2 and 4 were cultured with biotinylated MMPP (Bio-MMPP), both for 24 hours. We then performed a Streptavidin beads pull-down assay to assess the interaction between MMPP and CREB in cell lysates. The absence of CREB in lanes treated with non-biotinylated MMPP confirms the specificity of the interaction between biotinylated MMPP and CREB.

To address your concern about the background of non-specific pull-downs, we have added a negative control where Streptavidin beads were incubated without biotinylated MMPP. As shown in Figure 3, no bands were detected in this condition, confirming the absence of non-specific binding between Streptavidin beads and CREB.

While we recognize the importance of using a defective MMPP as a negative control, we currently do not have a suitable defective MMPP available, and therefore cannot perform this experiment at this time.

Figure 3. Streptavidin beads do not interact with CREB. Schematic diagram illustrating the functional domains of CREB (top) and Streptavidin pull-down assay (bottom) examining the interactions between Streptavidin beads and purified CREB proteins containing full-

length (FL) CREB, Q1 domain, KID domain, Q2 domain, or bZIP domain with MMPP (400 μ g/mL) treatment.

- Finally, a conceptual point. In Figure 1 the authors show a distribution of MMP that is both in the cortex and in the medulla. The accumulation is much more robust in the cortex than in the medulla, and in both cases it is cytosolic. If the interaction occurs directly with CREB, however, the MMP must be reaching the nucleus. And indeed the scheme in figure 8 shows a nuclear localization. Do the authors see an accumulation in the nucleus? How would stoichiometrically be explained the inhibition of CREB, particularly relevant in the medulla were the cAMP effect is believed to be maximal in PKD?

Response: Thank you for your insightful thought. In Figure 1F, our immunohistochemistry assays did reveal nuclear localization of MMPP, despite more pronounced cytoplasmic staining. To further elucidate MMPP's subcellular distribution, we labeled MMPP with FITC and employed confocal microscopy, as detailed in Appendix Figure S6 of the revised manuscript (originally Figure S9). These images confirm that MMPP predominantly localizes within the nucleus and lysosomes, with some association also noted with mitochondria and minimal overlap with the ER, clearly demonstrating MMPP's capability to enter the nucleus. Additionally, in Figure 5H of the revised manuscript (originally Figure 6H), pull-down assays were performed after separating cytoplasmic and nuclear fractions in Bio-MMPP-treated cells. The results indicated a specific interaction between MMPP and endogenous CREB predominantly in the nuclear fraction, affirming that MMPP can actively engage with nuclear CREB.

We agree with your point regarding the significance of the medulla in the progression of ADPKD. Our MRI results, shown in Figure EV2 of the revised manuscript (originally Figure S3), demonstrate MMPP accumulation in both the medulla and cortex. Although accumulation is more substantial in the cortex, the significant and sustained presence in the medulla for at least 24 hours should sufficiently enable ROS scavenging and CREB inhibition across both areas, thereby delaying disease progression.

Referee #2 (Comments on Novelty/Model System for Author):

Sun et al report that that ultra small polyethylene glycol incorporated Mn²⁺ chelated MNP (MMPP) traverse the glomerular filtration barrier and accumulate in renal tubules. They inhibited in vitro, ex vivo cystogenesis, and in vivo cystogenesis by a dual effect of melanin, as a highly effective antioxidant and by directly binding to the bZIP DNA binding domain of CREB, leading to competitive inhibition of CREB's DNA binding ability and subsequent reduction in CREB target gene expression. In addition, they engineered an orally administrable formulation that was as effective as the intraperitoneal administration. The studies were well conducted. Many observations are novel and, if confirmed, likely to have a substantial impact on the field.

Response: We thank the reviewer for this positive appraisal of our work.

Referee #3 (Remarks for Author):

In this manuscript, Sun et al. demonstrate the utilization of melanin-like nanoparticles (MNPs) for treating ADPKD. The authors have synthesized an ultrasmall MNP (MMPP) that efficiently crosses the glomerular filtration barrier and accumulates in the renal tubules. Their findings suggest that MMPP can significantly delay the progression of ADPKD by reducing oxidative stress and inhibiting the cAMP-CREB pathway. The manuscript proposes a novel therapeutic approach for ADPKD and identifies new intracellular targets for MNPs. The manuscript could be further improved if the authors address the following aspects:

1. The manuscript's primary finding is that MMPP inhibits CREB transcriptional activity, in addition to its antioxidant effects. To reinforce this conclusion, the authors should consider conducting control experiments using established antioxidants for ADPKD treatment. This would more convincingly demonstrate the specific impact of MMPP on CREB inhibition.

Response: We are grateful to the reviewer for their valuable suggestions regarding the control studies on CREB transcriptional activity. To investigate the specific inhibition of CREB by MMPP, we conducted experiments with other well-known antioxidants, Sulforaphane (SFN) and N-acetylcysteine (NAC). SFN functions by activating the Nrf2 pathway, thus boosting the body's antioxidant defenses, while NAC acts to replenish glutathione levels, directly mitigating oxidative stress. The efficacy of these compounds in treating ADPKD has been previously established by our group and others (Lu et al., *Sci. Transl. Med.*, 2020; Moyses et al., *Nephrol. Dial. Transplant.*, 2016).

To distinguish the CREB inhibition mechanism of MMPP from its general ROS scavenging effects, we employed WT 9-12 cells, which are derived from cysts of human PKD patients (Loghman-Adham et al., *Am J Physiol Renal Physiol*, 2003). Notably, these cells exhibit elevated ROS levels compared to normal kidney collecting duct cells (Ishimoto et al., *Mol. Cell Biol.*, 2017). We conducted ChIP-qPCR assays on these cells treated with MMPP, SFN, or NAC. As shown in Figure 1, there was a marked decrease in p-CREB recruitment on

NR4A2 and *PCK* genes, two established CREB target genes, in MMPP-treated cells. In contrast, such changes were not observed in cells treated with either SFN or NAC. These findings indicate that the inhibition of CREB binding to DNA by MMPP occurs independently of its ROS scavenging activity, highlighting the specificity of MMPP in targeting CREB function.

These data have been included in the Supplementary Information as Appendix Figure S7 (Page 18, Line 456-472).

Figure 1. ROS does not affect melanin nanoparticles' inhibition on CREB transcriptional activity. ChIP-qPCR analysis of p-CREB occupancy on *NR4A2* (left) and *PCK* (right) genes in WT 9-12 cells treated with MMPP or antioxidants (n =3).

2. Increased oxidative stress and activation of the cAMP signaling pathway are also significant contributors to polycystic liver disease (PLD). However, since liver cysts originate from cholangiocytes, it is essential to investigate whether MMPP accumulates in cholangiocytes to validate its efficacy in treating PLD.

Response: We are grateful for the reviewer's insightful question regarding the accumulation of MMPP in cholangiocytes. To address this concern, we performed PEG and CK19 staining (a specific marker for cholangiocytes) on liver samples from MMPP-treated mice. Our results, as shown in Figure 2, indicated significant colocalization of PEG and CK19 in the livers of MMPP-treated mice compared to control samples. This data indicates that MMPP

preferentially accumulates in cholangiocytes, and slows cyst growth in PLD.

Figure 2. MMPP accumulates in cholangiocytes. PEG staining in liver sections. Sale bars, 100 μ m.

3. During the progression of ADPKD, tubular injury can further exacerbate the disease. Therefore, the authors need to evaluate the impact of MMPP treatment on kidney injury.

Response: We thank the reviewer for this suggestion. In PKD, *neutrophil gelatinase-associated lipocalin (Ngal)* has been identified as a suitable marker for evaluating injury (Warner et al., *J Am Soc Nephrol.*, 2016). Therefore, we conducted RT-PCR analysis to detect *Ngal* gene expression. As showed in Figure 3, the expression of *Ngal* remained low in saline- or MMPP-treated WT mice, but markedly increased in ADPKD mice. However, in MMPP-treated ADPKD mice, its expression significantly decreased. These findings suggest that MMPP reduces kidney injury in PKD mice.

These data have been included in the Supplementary Information section as Appendix Figure S3B (Page 10, Line 229-230).

Figure 3. MMPP treatment reduces kidney injury in ADPKD mice. RT-PCR analysis of *Ngal* in kidneys from the indicated groups (n = 3). Data presented as means ± SD. Two-way ANOVA with LSD test was used for statistical analysis.

4.MNPs originate from various sources and thus exhibit diverse biological effects. Could the type of melanin used in the preparation of MMPP influence the results?

Response: We thank the reviewer for this suggestion. To assess the impact of varying melanin sources, we synthesized melanin-based nanoparticles using natural melanin (derived from cuttlefish, hair, and black sesame) and melanin-like substances (such as polydopamine). Aligning with findings in the existing literature (Deng et al., *ACS Nano.*, 2019; Hong et al., *ACS Nano.*, 2023; Chu et al., *Biomaterials*, 2016; Zhou et al., *Nat. Commun.*, 2023), our experiments indicated that these nanoparticles were generally larger than 50 nm. This size exceeds the filtration threshold of the renal glomerulus, thereby limiting their effective passage and subsequent reach to the renal tubules. In contrast, the specifically synthesized ultra-small MMPP demonstrated the optimal size for renal filtration, allowing effective delivery to the target sites within the kidneys.

5.The authors should include the genes from clusters 1-6 in the supplementary information to provide comprehensive data access.

Response: We are grateful for the reviewer's suggestion. We have included a comprehensive list of genes from cluster 1-6 in the Supplementary Information/Source data section.

6.While oxidative stress is a key factor in PKD pathogenesis, antioxidant therapies have failed in ADPKD clinical trials. The authors should compare previous studies in the discussion section, elucidating how MMPP differs from previous studies and why it might be better suited for clinical translation.

Response: Thank you for your constructive feedback. Previous research, including our studies and work by other groups, has shown that antioxidants like sulforaphane, bardoxolone methyl, and N-acetylcysteine can delay the progression of PKD by scavenging reactive oxygen species (ROS) (Lu et al., *Sci. Transl. Med.*, 2020; Moyses et al., *Nephrol. Dial. Transplant.*, 2016). However, these antioxidant treatments have not been effective in clinical trials for ADPKD, likely because targeting a single pathway may not sufficiently improve kidney function, or due to adverse effects on other organs.

Our research indicates that MMPP's mechanism of action uniquely combines ROS scavenging with inhibition of CREB transcriptional activity. This dual-targeting strategy enhances MMPP's specificity and effectiveness against ADPKD. Additionally, MRI studies show that due to its physicochemical properties, MMPP selectively localizes to the kidneys and liver, which greatly reduces the potential for off-target effects in other organs. These distinctive features position MMPP as a promising candidate for ADPKD treatment. This comparison has been incorporated into the Discussion section of our manuscript (Page 20, Line 512-514).

9th Oct 2024

Dear Dr. Zhang,

Thank you for submitting your revised study. We have now received the reports from the two referees who evaluated your revised manuscript. As you will see from the reports below, they are satisfied with the revisions, and I will therefore be able to accept your manuscript once the following editorial issues will be addressed:

1/ Please address the remaining minor comment from referee #1.

2/ Manuscript text:

- Please remove the red text and only keep in track changes mode any new modification.
- Methods:
 - o Thank you for providing a reagents and tools table, please remove it from the manuscript, and upload it as a separate file (using our template that you can find in our author guideline <https://www.embopress.org/page/journal/14693178/authorguide#structuredmethods>).
 - o Cells: please include the origin of the cells, and indicate whether they were authenticated and tested for mycoplasma contamination.
 - o Antibodies: please provide dilutions/concentrations.
 - o Blinding: please include a statement on blinding, even if no blinding was done.
- Data availability: thank you for depositing your sequencing data. Please note that the dataset must be made public before acceptance of the manuscript.
- Please rename "Competing interests statement" to "Disclosure statement and competing interests" (<https://www.embopress.org/competing-interests>).
- Please remove the list of appendices from the manuscript text.

3/ Figures and Appendix:

- Please correct the callouts "Table S1 of the Supplementary Information" and "Table S2 of the Supplementary Information" to Appendix Table S1, etc.
- Supplementary information should be uploaded as a PDF file renamed "Appendix", and include a table of contents with page numbers.
- Please address the queries from our copy editors in the figure legends:
 1. Please note that the exact p values are not provided in the legends of figures 2c, e-f, i; 3d, f; 4e; 6i; EV 1a-d; EV 4c.
 2. Please indicate the statistical test used for data analysis in the legends of figures 3b; 4d.
 3. Please note that information related to n is missing in the legends of figures 5a-b.
 4. Although 'n' is provided, please describe the nature of entity for 'n' in the legends of figures 6i; EV 1a-d; EV 4b-d.
 5. Please note that the error bars are not defined in the legends of figures 5a-b; EV 3a, c-e.
 6. Please note that the scale bar needs to be defined for figures EV 4a.

4/ Source Data:

Please upload the Source Data as one (zipped) file per figure. Please clarify to which figure the excel file "genes cluster 1-6" relates to.

4/ Checklist:

- Please clarify whether there are any restrictions on newly created materials.
- Please check whether you need to fill in the section "Cell materials/primary cultures".
- Please fill in the section "Cell materials/cell authentication and mycoplasma contamination".
- Please check the section "Experimental animals/animal observed in or captured from the field" as I don't think it applies to your study.
- Please check the section "Core facilities".
- Please check the section "Design/ protocol pre-registered".
- Please fill in the section "Experimental study design and statistic/inclusion/exclusion criteria".

5/ I introduced minor edits in your Paper Explained, please let me know if you agree with the following or amend as you see fit:
"Problem

Autosomal Dominant Polycystic Kidney Disease (ADPKD) is the most prevalent genetic kidney disorder globally, affecting approximately 12.5 million people. About 50% of individuals diagnosed with ADPKD are likely to progress to end-stage renal disease by their 50s or 60s. ADPKD involves multiple complex pathways, and treatments that target only one pathway often fail to effectively halt disease progression. Therefore, a significant challenge in the field is to develop safe, long-term treatments that can simultaneously target several key pathological pathways involved in ADPKD.

Results

In this study, we demonstrated that ultra-small polyethylene glycol-incorporated Mn²⁺-chelated melanin-like nanoparticles (MMPP) effectively inhibit cyst growth in an orthologous ADPKD mouse model. Upon systemic injection, MMPP crosses the glomerular filtration barrier and specifically accumulates in renal tubules. Beyond its classical antioxidant properties, we identified the cAMP-response element-binding protein (CREB) as a key intracellular target of MMPP. By directly binding to the bZIP domain of CREB, MMPP inhibits its transcriptional activity and the expression of downstream genes. Thus, MMPP targets both oxidative stress and CREB, two key pathways implicated in cystogenesis, leading to effective treatment of ADPKD in mice.

Impact

Our study enhances the therapeutic potential of melanin-like nanoparticles (MNPs) by uncovering specific intracellular targets and identifying new indications, such as ADPKD. This expands their applicability beyond chronic diseases associated with oxidative stress, to conditions involving dysregulated cAMP-CREB signaling. Our findings deepen our understanding of MNPs' mechanisms of action, paving the way for the development of safer and more effective melanin-based nanomedicines."

6/ Synopsis:

I introduced minor changes in your synopsis text, please let us know if you agree with the following or amend as you see fit:

"MMPP was shown to cross the glomerular filtration barrier and to accumulate in renal tubules, leading to reduced oxidative stress and inhibition of CREB transactivation. This dual action resulted in reduced cyst growth and improved kidney function in ADPKD.

- Systemically administered MMPP crossed the glomerular filtration barrier and selectively accumulated in the liver and kidneys.
- Oxidative stress was effectively reduced by MMPP treatment in vitro and in vivo.
- The transcription factor CREB was directly bound by MMPP, resulting in inhibition of the cAMP-CREB pathway and a reduction in CREB target gene expression.
- ADPKD disease progression was improved by MMPP via dual inhibition of oxidative stress and CREB transcriptional activity.

Thank you for providing a nice synopsis image. Please resize it to 550 px wide x 300-600 px high and make sure the text remains legible. I cropped a small portion (115x70 pixels) that will serve as thumbnail for the table of content on our webpage (attached). Please let us know if you agree, or provide a different thumbnail with the same dimensions as changes during proofing are usually not allowed.

8/ As part of the EMBO Publications transparent editorial process initiative (see our Editorial at <http://embomolmed.embopress.org/content/2/9/329>), EMBO Molecular Medicine will publish online a Review Process File (RPF) to accompany accepted manuscripts.

This file will be published in conjunction with your paper and will include the anonymous referee reports, your point-by-point response and all pertinent correspondence relating to the manuscript. Let us know whether you agree with the publication of the RPF and as here, if you want to remove or not any figures from it prior to publication.

I look forward to receiving your revised manuscript.

Yours sincerely,

Lise Roth

***** Reviewer's comments *****

Referee #1 (Comments on Novelty/Model System for Author):

The investigators have carried out the most convincing part of their studies in appropriate murine models of the disease.

Referee #1 (Remarks for Author):

The investigators have responded to all the requests of this reviewer. The only minor point that remains open is that in Figure 2 they now indicate the dot that corresponds to the shown representative images, but a black dot in a dark blue graph is difficult to visualize. I would suggest using a different color, or an arrow to clearly identify the sample that is being shown.

Referee #3 (Remarks for Author):

The authors have addressed all comments and revised the manuscript. It is acceptable for publication.

Referee #1 (Comments on Novelty/Model System for Author):

The investigators have carried out the most convincing part of their studies in appropriate murine models of the disease.

Referee #1 (Remarks for Author):

The investigators have responded to all the requests of this reviewer. The only minor point that remains open is that in Figure 2 they now indicate the dot that corresponds to the shown representative images, but a black dot in a dark blue graph is difficult to visualize. I would suggest using a different color, or an arrow to clearly identify the sample that is being shown.

Response: Thank you to the reviewer for the positive feedback and for highlighting the issue with the label in Figure 2. In the revised version, we have used arrows to clearly mark the representative samples shown in the graph (Figure 2C, E, and I).

18th Oct 2024

Dear Dr. Zhang,

Thank you for submitting your revised files. I am pleased to inform you that your manuscript is accepted for publication and is now being sent to our publisher to be included in the next available issue of EMBO Molecular Medicine.

If you have any questions, please do not hesitate to contact the Editorial Office.

Thank you for your nice contribution to EMBO Molecular Medicine!

Yours sincerely,

Lise Roth
